# Variable-resolution building exposure modelling for earthquake and tsunami scenario-based risk assessment. An application case in Lima, Peru

Juan Camilo Gomez-Zapata[,1,2], Nils Brinckmann[3], Sven Harig[4], Raquel Zafrir[1,5], Massimiliano Pittore[1,6], Fabrice Cotton[1,2], Andrey Babeyko[7]

[1] Seismic Hazard and Risk Dynamics, GFZ German Research Centre for Geosciences, Potsdam, 14473, Germany

[2] Institute for Geosciences, University of Potsdam, Potsdam, 14469, Germany

[3] eScience Centre, GFZ German Research Centre for Geosciences, 14473, Potsdam, Germany

[4] Alfred Wegener Institute, Helmholtz Centre for Polar and Marine Research (AWI), Bremerhaven, 27570, Germany

[5] Head Aerospace Group, Paris, 92250, France

[6] Institute for Earth Observation, EURAC Research, 39100, Bolzano, Italy

[7] Geodynamic Modelling, GFZ German Research Centre for Geosciences, 14473, Potsdam, Germany

*Correspondence to*: Juan Camilo Gomez-Zapata (jcgomez@gfz-potsdam.de)

**Abstract.** We propose the use of variable resolution boundaries based on Central Voronoi Tessellations (CVT) to spatially aggregate building exposure models for risk assessment to various natural hazards. Such a framework is especially beneficial when the spatial distribution of the considered hazards present intensity measures with contrasting footprints and spatial correlations, such as in coastal environments. This work avoids the incorrect assumption that a single intensity value from hazards with low spatial correlation (e.g., tsunami) can be considered to be representative within large-sized geocells for physical vulnerability assessment, without, at the same time, increasing the complexity of the overall model. We present decoupled earthquake and tsunami scenario-based risk estimates for the residential building stock of Lima (Peru). We observe that earthquake loss models for far-field subduction sources are practically insensitive to the exposure resolution. Conversely, tsunami loss models and associated uncertainties depend on the spatial correlations of the hazard intensities as well as on the resolution of the exposure models. We note that for the portfolio located in the coastal area exposed to both perils in Lima, the ground-shaking dominates the losses for lower magnitude earthquakes, whilst tsunamis cause the most damage for larger magnitude events. For the latter, two sets of existing empirical flow-depth fragility models are used, resulting in large differences in the calculated losses. This study, therefore, raises awareness about the uncertainties associated with the selection of fragility models and spatial aggregation entities for exposure modelling and loss mapping.

## 1 Introduction

The spatial distribution of damage and/or losses expected to be incurred by an extensive building portfolio from a natural hazardous event can be quantified and mapped once a physical vulnerability analysis is performed. For such a purpose, a set of fragility functions per building class is conventionally used. Fragility functions describe the probability of exceeding a certain damage limit state for a given intensity measure (IM) associated with a natural hazard, such as spectral acceleration at the yield period (e.g., Fäh et al., 2001) for earthquakes, or tsunami inundation height for tsunamis (Koshimura et al., 2009). These vulnerability calculations are performed at the centroid of every aggregation unit of a building exposure model with some level of uncertainty associated

with them (Bazzurro and Luco, 2005), or over weighted locations (e.g., Weatherill et al., 2015; Kappos et al., 2008). These aggregation entities can be very diverse, ranging from administrative units such as district/communes (e.g., Dunand and Gueguen, 2012), urban blocks (e.g., Papathoma and Dominey-Howes, 2003; Kappos et al., 2008; Figueiredo et al., 2018; Kohrangi et al., 2021), regular grids (e.g., Erdik and Fahjan, 2008; Figueiredo and Martina, 2016) or variable-resolution CVT (Central Voronoi Tessellation) geocells (Pittore et al., 2020). Throughout the physical vulnerability assessment, it is implicitly assumed that the

intensity observed or estimated at that location (i.e., centroid or weighted points) is representative for the entire aggregation area. Depending on the considered hazard footprint and IM attenuation, this assumption might not be valid if the aggregation area is too coarse compared to the correlation length of a highly varying IM. In addition to the aggregation of the building exposure itself, the importance of these geographical aggregation entities in natural hazard risk assessment is that they are ultimately used to calculate and map the expected damage and loss metrics (e.g., building replacement/repair costs, human casualties). The diverse types of

visualisation and interpretations of this kind of geospatial data define the so-called thematic uncertainties (Smith Mason et al., 2017) that can heavily impact upon the decision making processes (Viard et al., 2011). It is, therefore, important to find a compromise between the intrinsic resolution of the hazard IM, on the one hand, and the cartographic representation of the exposure models and risk metrics on the other.

     When a geographically distributed hazard IM presents no significant spatial variability within distances of the order of

tens of kilometres, they are said to be highly spatially correlated (e.g., Gill and Malamud, 2014; Merz et al., 2020). This is the case of hazards with relatively low attenuation and wide-spread footprints, such as ash-falls and earthquakes (de Ruiter et al., 2021). For the latter case, when seismic site conditions (e.g., soil amplification) and path effects (e.g., seismic directivity) are insignificant, seismic ground motion correlation lengths between 10 km and 25 km are typical (e.g., Esposito and Iervolino, 2012; Schiappapietra and Douglas, 2020). On the other hand, hazards are described as being low-spatially correlated if their IM are highly prone to be

modified by specific features of the propagation medium. For instance, the modelling of inland IMs from a tsunami (i.e., inundation depth, flow velocity, momentum flux) are highly dependent on the nature and resolution of the bathymetry and digital elevation models (e.g., Tang et al., 2009), coastal topography (e.g., Goda et al., 2015), coastal morphology (e.g., Song and Goda, 2019), and even the nature of the built-up areas that have the potential to interact with and modify the inundation footprint and flow velocities (e.g., Kaiser et al., 2011; Lynett, 2016). Moreover, in the case of earthquake-triggered tsunamis, the maximum tsunami IMs also

depend on the properties of the triggering mechanism, for example, the earthquake's magnitude (e.g., Goda et al., 2014), slip distribution (Miyashita et al., 2020), and directivity of the radiated energy (e.g., Kajiura, 1972). Thus, the spatial correlation of inland IMs from tsunamis is very low and remarkably non-linear compared to the much more uniform and highly spatially correlated seismic ground motion. Efforts to visualize uncertainties in the tsunami hazard and risk mapping that address some of the aforementioned modifiers have been reported in a few studies (e.g., Goda and Song, 2016; Goda et al., 2020).

Usually, the resolution of exposure models is constrained independently of the hazard, and to a certain extent, also independently of the exposure distribution. That might lead to poor exposure resolutions where it really matters, i.e., in areas where buildings are densely distributed and/or hazard intensities vary over short distances. Or, by contrast, to the unnecessary computation demands for loss assessment in areas with few exposed assets. If aggregation areas of the exposure model are coarser in resolution than the typical correlation lengths of low spatially correlated hazard intensities, then local variations in these intensities would

remain hidden in the vulnerability analysis, propagating the associated uncertainties up to the loss estimates. This dependency between exposure resolution and spatial correlation of hazard intensities has been usually disregarded, although some examples can be found in soil liquefaction risk assessment. Despite the hazard component can be spatially downscaled (e.g. Bozzoni et al., 2020), thematic uncertainties related to visualisation and the interpretation of risk metrics can arise if they are mapped over larger regional administrative units (e.g., Yilmaz et al., 2021) instead of being represented at more hazard-compliant resolutions (e.g.,

Bozzoni et al., 2021). Similarly, despite building exposure models for flood and earthquake vulnerabilities being able to be aggregated at moderate resolutions (e.g., 4x4 km grid in Dabbeek and Silva, 2019), similar thematic uncertainties can evolve due to the profound differences between both spatially correlated hazard intensities, and when the calculated losses are mapped over regional administrative units (Dabbeek et al., 2020).

     To the best of the authors' knowledge, consideration of different hazard footprints and the spatial correlation of their
intensity measures for the construction of aggregation entities for exposure modelling has been seldomly discussed in the literature. For instance, Chen et al. (2004) described the importance of ensuring a consistent delimitation of the resolution of exposure models along with the spatial variation of their two considered hazards, earthquakes and hailstorms, that impose damage footprints of very different extents. Meanwhile, Douglas (2007) and Ordaz et al. (2019) highlighted the importance of the geographical scale to represent the building exposure models that are affected differently, depending on variable hazard footprints. The study reported
in Zuccaro et al. (2018) is perhaps the most advanced framework in the state of the art for the construction of a common aggregation entity for multi-hazard risk assessment, referred to as the minimum reference unit (MRU). This geographical unit coincides with the minimum resolution of analysis of input (i.e., hazard intensities and exposure model) and output elements (i.e., damage and loss estimates) and remarks that despite high-resolution hazard models, one would achieve neither an accurate risk assessment nor meaningful loss mapping if there is no compatibility between the cartographic representation of the building exposure model, the
hazard footprints, and their IM correlation.

     A denser set of geocells in the same area occupied by a coarser regular-sized cell or administrative units provides a denser arrangement of hazard intensity values (when there are local IM variations) to the set of fragility functions per considered hazard. When local IM variations are not sufficiently represented into finer aggregation entities during the vulnerability analysis, thematic uncertainties might appear in the mapping, visualisation, and interpretation of the loss estimates. Therefore, besides the
conventional epistemic and aleatory uncertainties linked to the hazard, exposure, and vulnerability components, thematic uncertainties are also present in the risk chain when the loss metrics are mapped. Awareness of the thematic uncertainties as well as clear and meaningful vulnerability/loss mappings towards the most relevant hazards a community is exposed to is necessary to improve urban planning, mitigation strategies, and emergency response actions (e.g., Pang, 2008; Aguirre-Ayerbe et al., 2018).

     We can distinguish two types of approaches formerly proposed in the literature to investigate the exposure aggregation
for natural hazard risk applications.

**(1)** To independently represent the building portfolio over a series of aggregation entities such as administrative units, or equidimensional regular grids, and explore their individual contribution to the uncertainty in the losses imposed by certain hazard(s) to ultimately select a representative aggregation model. This option has been explored in Bal et al. (2010), Frolova et al. (2017), Senouci et al. (2018), and Kalakonas et al. (2020) for seismic vulnerability applications, and in Figueiredo and
Martina (2016) for flood vulnerability. These studies discuss the weakness of physical vulnerability mapping at the individual building scale and over coarse aggregation areas and highlight the importance of finding an optimal resolution for building exposure modelling while minimizing the uncertainties in the loss estimates. However, these attempts did not explicitly address the spatial correlation or attenuation of the hazard intensity onto the predefined aggregation areas and focused on the vulnerability towards individual hazards rather than on multi-hazard risk applications.

**(2)** Aggregating the exposure models over variably resolved entities that are not necessarily administrative boundaries. This has been done in fewer studies. For instance, Muis et al. (2016) assessed the global population exposure to coastal flooding (from storm surges and extreme sea levels) through the application of a hydrodynamical model based on unstructured grids to ensure sufficient resolution in shallow coastal areas. Scheingraber and Käser (2019) explored the uncertainty in regional building portfolio locations for seismic risk through the use of weighted irregular grids. This weighting was carried out as a function

of the population density and did not use any hazard IM or footprints. Scheingraber and Käser (2020) described the advantages of the former procedure in terms of computational efficiency and the treatment and communication of uncertainties in probabilistic seismic risk assessment on a regional scale. Alternatively, aggregating the building portfolio into anisotropic CVT-based geocells (Central Voronoidal Tessellations) is suggested by Pittore et al. (2020).

In this study, we employ anisotropic CVTs to aggregate the residential building exposure models. Voronoi regions have proved to be useful in geographical partitioning (e.g., political districting, Ricca et al., 2013), as well in other hazard-related applications, such as climatological modelling (e.g., Zarzycki and Jablonowski, 2014). We present for the first time how CVT can be constructed using underlying combinations of geospatial distributions to achieve a larger resolution of spatially aggregated building portfolios where it matters for risk assessment. We adapt and customize their derivation to explicitly account for the combination of a low-correlated hazard intensity (tsunami inundation) and one exposure proxy (population density) to generate the Voronoi regions.

The aggregated building portfolios are used for earthquake and tsunami scenario-based risk applications. We have systematically investigated six megathrust subduction earthquakes and their respective tsunamis with moment magnitudes ranging between 8.5 and 9.0. We consider the residential building stock of Metropolitan Lima (Peru) classified in terms of one set of earthquake vulnerability classes and two sets of tsunami vulnerability classes. These building portfolios have been aggregated within six customized CVT models and administrative units at the highest resolution available (i.e., the block level). By using the respective set of fragility functions, we have independently calculated the direct losses from scenario-based physical vulnerability analyses (seismic ground-shaking and tsunami inundation). We show that the implementation of this approach is beneficial not only in finding a balance between accuracy and computational demand, but also in the efficient representation of the loss estimates while reducing bias generated in the loss mapping. The role of the spatial correlation of both hazard intensities in the efficiency and accuracy of the CVT-based exposure models is also discussed. Since the main scope of this work is to investigate an efficient manner to aggregate the building exposure for risk applications considering multiple hazards, we have not investigated the conditional probabilities related to cascading events (e.g., Goda et al., 2018). Instead, we have assumed that every seismic rupture produces a tsunami. Hence, we are not accounting for cumulative damage on buildings due to consecutive ground shaking and tsunami (e.g., Park et al., 2019; Negulescu et al., 2020; Goda et al., 2021) nor the risk to other seismically induced hazards (i.e., earthquake-triggered landslides, liquefaction, ground failure, etc., e.g., see Daniell et al., 2017).

## 2    Methodology to construct variable-resolution exposure model for risk assessment to multiple hazards.

The proposed methodology is composed of the following steps:

1.  Simulation of scenario-based hazards (i.e., earthquakes and tsunamis) with the same spatially distributed intensities required by each fragility assessment.

2.  Construction of one (or a set of) representative underlying spatial distributions (i.e., focus map(s) See Sect. 2.2). This implies the selection and ranking (with numerical weights) of the hazard intensities or exposure proxies.

3.  Generation of CVT-based aggregation entities employing the focus map as an underlying distribution and with different numbers of seeding points.

4.  Classification of the exposed building stock of interest into vulnerability classes per considered hazard and their aggregation into the CVT-based geographical entities.

5.  Scenario-based risk assessment independently per hazard type and discussion of their associated thematic uncertainties in the loss mapping and visualisation.

The uncertainties arising from steps 3 and 5 are explored through the formulation of a condition tree.

## 2.1 Simulation of scenario-based hazards with spatially distributed intensities

We employ numerical earthquake and tsunami scenarios to replicate historical or hypothetical future events to simulate spatially distributed hazard intensities. For earthquakes, we simulate ground motions from suitable GMPE (ground motion prediction equations). Cross-correlated ground motions are generated for the spectral-periods that serve the fragility functions as intensity measures (IM). For tsunamis, we employ the physical generation and propagation model TsunAWI (Harig et al., 2008) and simulate coastal inundation as the IM for tsunamis. The spatially distributed tsunami intensity values (inundation flow depth) are compatible with the IM of the selected set of fragility functions required in the vulnerability analysis.

## 2.2 Construction of focus maps

The focus map drives the construction of a variable-resolution exposure model for aggregating building portfolios. Focus maps were first introduced by Pittore (2015) based on the work of Dilley (2005), who proposed the spatial representation of composite indicators in hot-spots. Eq. (1) recalls the definition of a focus map, $S(D_i)$, that represents the probability of each location to be highlighted, given the actual values of certain indicators $D_i$.

$$S(D_i) = P(S|D_i) \in [0,1] \tag{1}$$

By using a pooling operator, a focus map highlights areas where a weighted combination of various normalized spatially distributed indicators ($D_i$ jointly assume the larger values. We propose to obtain a focus map that drives the aggregation entities for earthquake and tsunami exposure modelling through the combination of two indicators. (1) Population density ($D_0$ (from aggregated data sources e.g., WorldPop; GPWv4 (CIESIN, 2018)). This indicator is an exposure proxy about the location of residential buildings for which their ground-shaking vulnerability should be addressed. The use of the latter can be a useful indicator when other seismic risk components such as soil amplification conditions are poorly known, come from proxies with coarse resolutions (e.g., topography-based), or when strong seismic site effects are not expected. (2) The tsunami component is constrained through the expected tsunami inundation height ($D_1$ obtained from a "worst-case scenario" approach (i.e., largest feasible intensities) among a series of deterministic scenarios (e.g., Omira et al., 2009; Wronna et al., 2015). For the combination of the two aforementioned normalized input layers, we use a log-linear pooling operator $P_G$, as outlined in Eq. (2). This algorithm assigns a higher sampling probability to spatial locations where both indicators are relevant while penalizing the locations where at least one of the indicators (i.e., tsunami) is negligible.

$$lnP_G\big(P(S|D_0), P(S|D_1), .. P(S|D_n)\big) = lnZ + \sum_{i=0}^{n} w_i lnP(S|D_i) \tag{2}$$

where $Z$ is a normalizing constant and $w_i$ represents the respective weight assigned to score the relevance of each input layer, and $\sum w_i = 1, w_i > 0 \forall i$. Thus, the construction of a focus map entails the selection of the weights that rank the importance of every layer, as such carries its own epistemic uncertainties.

## 2.3 Generation of CVT-based exposure models

Selectively increasing the spatial resolution of aggregated areas is beneficial for capturing low spatially correlated hazard intensity values such as tsunami inundation heights. This is achieved by the construction of geocells with variable resolution in the form of CVT. During this construction, the focus maps are used as underlying spatial intensities and are sampled using a Poisson point process (Cox and Isham, 1980) to generate a number of seeding points. These points are used as centroids of the Voronoi geocells and through an iterative relaxation process will converge to the final CVT geocells. The number of seeding points therefore defines

the number of geocells of the resulting tessellation. CVTs are computed in various iteration steps using the simple relaxation method originally proposed by Lloyd (1982) until the distance between the geometrical centroid of the geocell and the weighted mass centroid generated by the raster distribution falls below a defined threshold, or after a given maximum number of iterations. Since the relaxation process is based on the underlying focus maps as generating distribution, CVT cell sizes are inversely proportional to the intensity of the focus map. Each CVT geocell in fact becomes a minimum resolution unit, as proposed in Zuccaro et al. (2018), and the resulting tessellation sets the basis for a variable-resolution exposure model. Voronoi regions inherently fulfil spatial properties such as compactness and contiguity (without holes or isolated parts) (Ricca et al., 2008).

### 2.3.1 Condition tree for multi-hazard exposure modelling

The epistemic uncertainties underlying the two steps discussed above are explored using a condition tree with the following hierarchical levels:

  i.   Selection of a suitable scheme (sets of building classes and their associated fragility functions) to describe the building inventory in the study area for risk assessment.

  ii.  Weight arrangement values ($w_i$ that rank every input layer (low spatially correlated hazard intensities or spatial proxies related to the exposure component) in the focus map construction.

  iii. Determination of the number of seeding points sampling the Poisson Point Process driven by a focus map that drives the generation of CVT-based geocells.

The condition tree presents a summary of assumptions for uncertainty treatment (Beven et al., 2018). Through the construction of alternative multi-resolution exposure models, the impact of every level of the condition tree is systematically investigated once the vulnerability assessment is performed.

## 2.4 The classification of the building stock into vulnerability classes and aggregation

The building stock of interest is classified in terms of several sets of mutually exclusive, collectively exhaustive vulnerability classes, whose aggrupation describe a set of classes (scheme) specific to the considered hazard (i.e., earthquake and tsunami). A top-down approach is used to make use of aggregated census data and ancillary data for the seismic-oriented building classes. Subsequently, the proportions assigned to each seismic-oriented building classes are reassigned to tsunami oriented classes through the use of inter-scheme compatibility matrices as presented in Gomez-Zapata et al. (2021). This method is partly based on the taxonomic disaggregation of every building type within a source and a target scheme as proposed by Pittore et al. (2018) for seismic vulnerability applications, and by Charvet et al. (2017) for the definition of tsunami-oriented building classes. Then, the classified building stocks are aggregated into every CVT model obtained in the former step.

## 2.5 Scenario-based risk assessment

The fragility of the building portfolio to the considered earthquakes and tsunami scenarios is calculated separately over every aggregation exposure model (see Sect. 2.1). This decision is based on the recent findings of Petrone et al. (2020) who found fundamentally different structural responses to both perils. Consequently, the authors argued that the intensity of the seismic ground motion does not play a significant role in the building's structural tsunami response unless it induces structural yield. The latter is assumed for the vulnerability analysis, considering the objective of this study of evaluating an optimal exposure model for risk assessment from the considered hazards. The scenario-based risk assessment makes use of a set fragility function associated with each building class and whose IMs are compatible with the hazard intensities modelled in Sect. 2.1. Each of their damage states has an assigned loss ratio to total replacement cost per hazard-dependent building class.

# 3    Application example

## 3.1  Context of the study area: Metropolitan Lima, Peru

According to Petersen et al., (2018), Peru, among all the South American countries, has the largest number of inhabitants, and considering a 10% probability of exceedance in 50 years, may experience a ground-shaking greater than VIII (modified Mercalli intensity scale, MMI). This makes Peru the country in which the largest average annual fatalities from earthquakes are expected in South America. In the same study, Lima, with nearly 10 million inhabitants, has been identified as the capital city exposed to the highest seismic hazard in the region, in line with Salgado-Gálvez et al. (2018), who also ranked the city's seismic hazard at the same level. Earthquakes are mostly attributed to the oceanic Nazca Plate subducting beneath the South American plate (Villegas-Lanza et al., 2016). In 1746, a giant megathrust earthquake with an estimated magnitude of Mw 8.8 (Jimenez et al., 2013) ruptured along some 350 km and triggered a tsunami with local run-up heights of 15 to 20 m (Dorbath et al., 1990), destroying the cities of Lima and Callao. In addition, the less studied events of 1586 and 1724 triggered tsunami run-ups of over 24 m (Kulikov et al., 2005). Moreover, according to Løvholt et al. (2014), Peru has the largest population exposed to tsunami hazard in the American continent, with Lima representing around 1/3 of the total country's population. In addition, the study area hosts the most important economic activities, as well as the main port of the country. Consequently, in Schelske et al. (2014), Lima has been ranked as the second metropolitan area in the world in terms of the value of working days lost relative to the national economy due to earthquakes. This highlights the relevance of integrated vulnerability studies in this study area for the overall economy of Peru.

Local authorities have conducted studies for emergency management and recovery planning considering tsunami and earthquake scenarios (e.g., PREDES, 2009), including qualitative risk estimations. Among others, the Japanese SATREPS project contributed to the improvement of the exposure model of Lima using satellite imagery and census data (Matsuoka et al., 2013). On the seismic vulnerability side, the project SARA, led by the Global Earthquake Model (GEM), made a significant contribution to classifying the residential building stock of Peru (Yepes-Estrada et al., 2017) and to assign appropriate fragility functions for similar classes (Villar-Vega et al., 2017), while more specific models for confined masonry have been reported in Lovon et al., (2018). On the tsunami vulnerability side, Adriano et al., (2014) estimated tsunami damage probabilities for two tsunami scenarios over the residential building portfolio classified into four building classes and employed the empirical tsunami fragility functions developed by Suppasri et al. (2013) after the 2011 Japan tsunami. Furthermore, Ordaz et al., (2019) analysed the probabilistic physical risk to both hazards in Callao, remarking upon the importance of addressing simultaneous losses.

## 3.2  Construction of earthquake and tsunami scenarios for Lima

We have simulated six earthquakes and tsunami scenarios offshore of Peru with moment magnitudes between Mw 8.5 to 9.0. Finite fault ruptures are modelled using the OpenQuake engine (Pagani et al., 2014) emulating the historical earthquake that occurred in 1746, in line with previous studies (e.g., Mas et al., 2014; Pulido et al., 2015; Ceferino et al., 2018a). The basic parameters used in the simulations are hypocentre location (longitude = -77.93°; latitude = -12.19°; depth = 8 km), strike = 329°, dip = 20°, and rake = 90°. Spatially distributed ground motion fields (GMF) were generated using the GMPE proposed by Montalva et al. (2017). Its site term is based on the shear wave velocity in the uppermost 30 meters depth ($Vs_{30}$) as reported in Ceferino et al. (2018b) in which the slope-based $Vs_{30}$ values (Allen and Wald, 2007) and seismic microzonation (Aguilar et al., 2013) were compiled and merged to the same resolution (30 arc-seconds ~ 1 km). The aleatory uncertainty in the ground motions was addressed by generating 1,000 realisations per event, as advised in Silva (2016), with uncorrelated and cross-correlated ground motion residuals. For the latter case, we used the Markhvida et al. (2018) model for PGA, and spectral acceleration for periods 0.3 s and 1.0 s. Examples considering three magnitudes (Mw 8.6, 8.8, and 9.0) and the respective tsunami scenarios are shown in Figure 1.

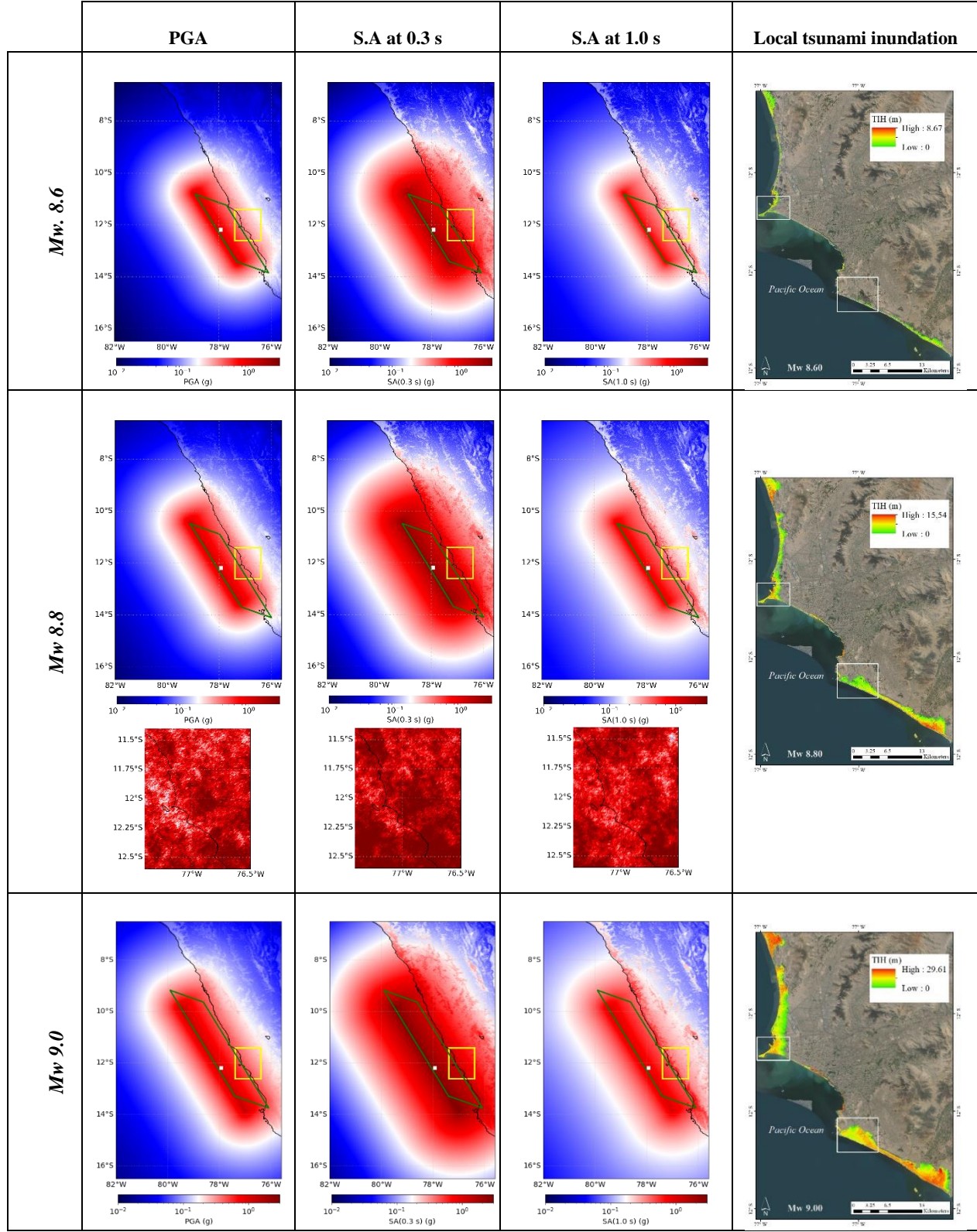

**Figure 1**. Median seismic ground motion for a single realisation using the Montalva et al. (2017) GMPE for PGA and spectral acceleration for periods 0.3 s and 1.0 s, and for three earthquake scenarios (Mw 8.6, 8.8 and 9.0) along the Peruvian subduction zone. Green rectangles represent the rupture planes. Hypocentres are shown by white dots. The study area (Metropolitan Lima) is enclosed within a yellow rectangle. For this area, and for the Mw 8.8 scenario, there is shown one realisation of spatially cross-correlated ground motion field per spectral acceleration. Tsunami inundation heights for the three selected scenarios are displayed for the study area. The northern "La Punta" sector (Callao district) and the southern Chorrillos district are indicated by white rectangles. Map data: ©Google Earth 2021.

Although a sensitivity analysis on the GMPE(s) selection is outside the scope of this study, such a choice may influence the resulting cross-correlated ground motion fields. This comes from the manner in which the residuals and soil nonlinearity are accounted for in the functional form of the selected attenuation model (Weatherill et al., 2015). Although the Montalva et al. (2017) GMPE uses $Vs_{30}$ as the site exploratory variable and includes nonlinear site response, the spatial resolution of the geo-dataset we have used might be too coarse to capture local variability in ground motion. These features could only be approximated through site-response analyses that account for the local geotechnical soil properties of site-specific soil profiles, as for instance performed by Aguilar et al., (2019) after applying the equivalent-linear methodology.

Tsunami simulations are based on the source parameters suggested by Jimenez et al. (2013). We fixed all earthquake parameters except for the slip value specifying the magnitude over a range from Mw 8.5 to Mw 9.0. This simplifies the simulation process and allows for a more systematic study of the contribution of the event´s magnitude and the corresponding tsunami footprint upon the loss assessment for the aggregated building exposure models. The wave propagation and tsunami inundations are obtained through numerical simulations using the finite element model TsunAWI which employs a triangular mesh with variable resolution, allowing for a flexible way to discretize the model domain with good representation of coastline and bathymetric features. Since the simulation of the inundation process needs high resolution, the mean mesh resolution given by the triangle edge length amounts to around 20m in the coastal area of Lima and Callao. TsunAWI is based on the nonlinear shallow water equations including parameterisations for bottom friction and viscosity. Table 1 summarizes some of the most important model quantities. The wetting and drying scheme is based on an extrapolation method projecting model quantities between the ocean part and the dry land part of the model domain, with the resulting simulations included in Harig and Rakowsky, (2021).

**Table 1.** Summary of TsunAWI model parameters used in the tsunami simulations.

| Numerical approach | Time step/ Integration time | Resolution range (Triangle edge length) | Bottom friction parameterization | Viscosity parameterization |
|---|---|---|---|---|
| Finite Elements | 0.1sec / 4 hrs | From 6km (deep ocean) to 7m (coastal pilot areas) | Manning (n=0.02 constant value) | Smagorinsky |

Figure 2 displays a small section of the model domain and shows the resolution of the triangular mesh which is directly connected to the water depth and bathymetry gradient in the ocean, whereas the edge lengths are shortest in the coastal part of the study area, where tsunami inundation is expected. The model bathymetry and topography were built from several data sets. The ocean part is based on the GEBCO bathymetry (General bathymetric chart of the ocean, GEBCO_08 Grid, version 20090202, see http://www.gebco.net). The coastal topography is from the SRTM topographic model (Shuttle radar topography mission, 30m resolution, see https://www2.jpl.nasa.gov/srtm/), whereas in the pilot area Lima/Callao, results from the TanDEM-X mission (Krieger et al., 2007) with a spatial resolution of 12 m were provided to the RIESGOS consortium. In this region, the available data sets were combined into a joint product and augmented by nautical charts of the shallow areas by the project partner EOMAP. All these data were bilinearly interpolated to the triangular mesh and slightly smoothed to allow for stable simulations. The raw model output in the triangular mesh as shown in Figure 2 contains all information at the model's resolution. However, it is cumbersome to process and visualize, since the data are given at unstructured locations and need statistical processing. Therefore, the results are interpolated to a raster file before providing them to project partners. Considering the mean resolution of the triangular mesh, a raster with grid cell dimensions of 10 x 10 m was chosen. An example of the resulting mesh and data product for the port area of Callao is shown in Figure 3. More details of this method are reported in Harig et al. (2020).


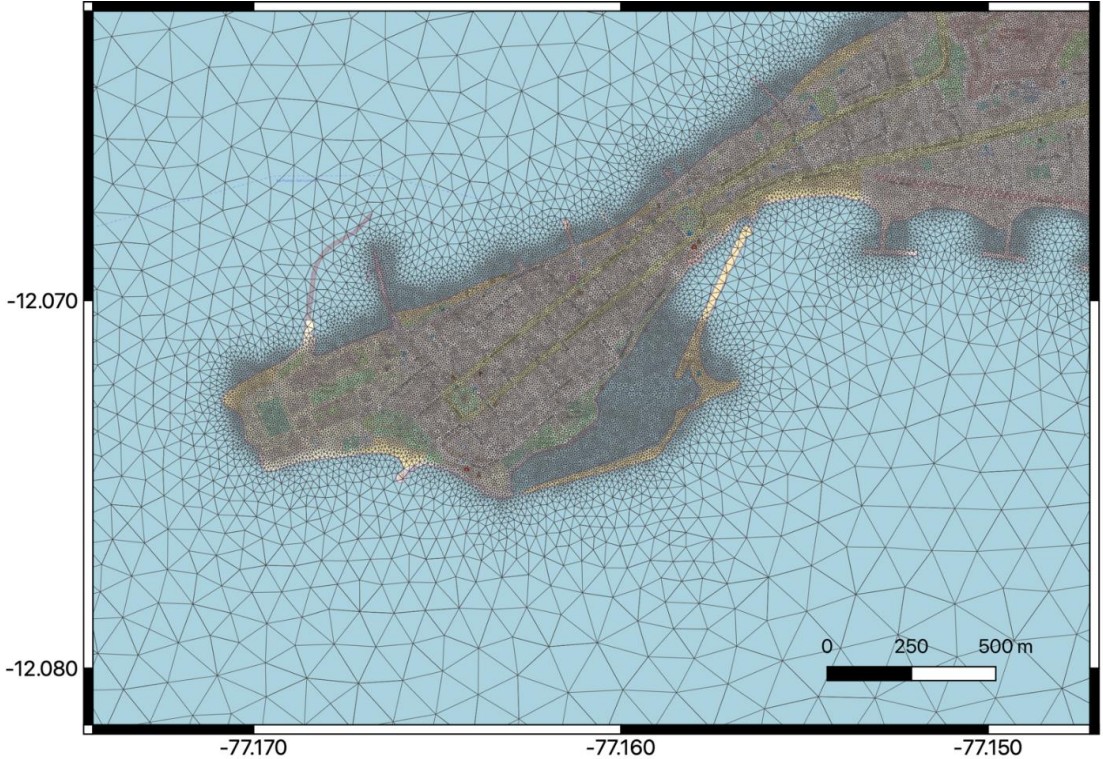

**Figure 2.** Section of the triangular mesh used for the TsunAWI simulations in the La Punta sector (Callao district). The mean resolution in the pilot area is approximately 20 m, whereas the shortest edge length measures about 7 m. The basemap and data are from © OpenStreetMap contributors 2021. Distributed under the Open Data Commons Open Database License (ODbL) v1.0.


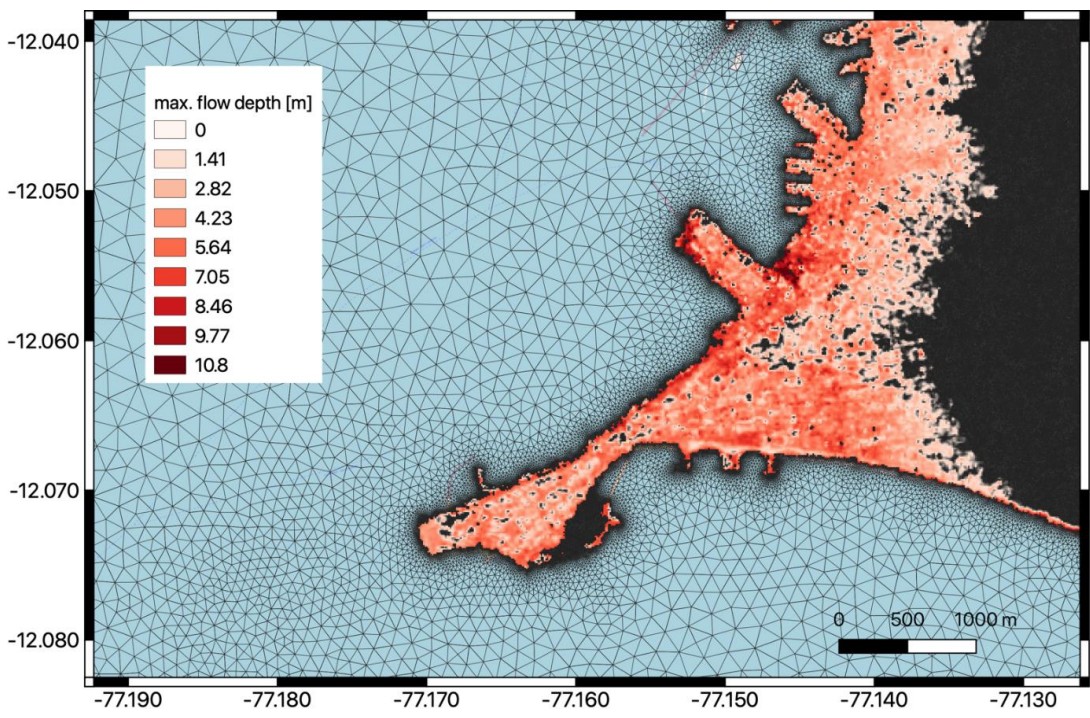

**Figure 3.** Section of the triangular mesh together with the inundation data product (10 m raster) for the tsunami scenario involving a magnitude 8.8 event in the Callao Harbour area. The basemap is from © OpenStreetMap contributors 2021. Distributed under the Open Data Commons Open Database License (ODbL) v1.0.

It is worth mentioning here that we do not aim to validate the inundation results for a specific event, which would require the optimization of the elevation and source model. Rather, we investigate the tsunami impact for a range of magnitudes with simplified sources. Tsunami inundation heights from the six scenarios over the two most tsunami-prone areas in Lima city and the La Punta and Chorrillos districts (see the white square in the tsunami maps of Figure 1) are shown in Figure 4. Conversely, significant tsunami inundation is not expected in the central Lima area due to the presence of sizable cliffs.

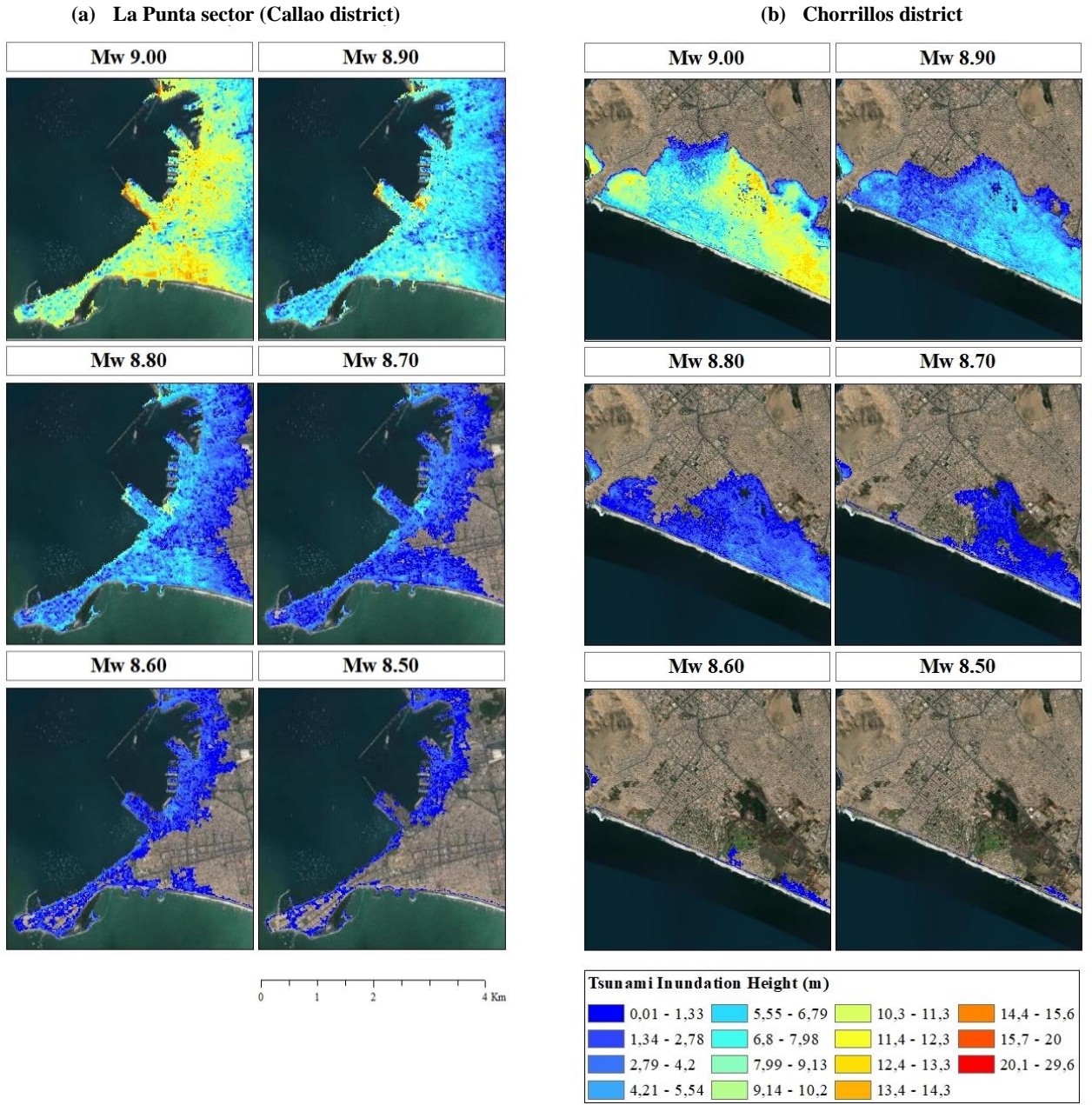

**Figure 4.** Expected tsunami inundation height (m) for two local areas within Lima for six tsunami scenarios (with locations in Figure 1). Map data: ©Google Earth 2021.

### 3.3 Construction of the focus maps

Focus maps have been constructed as inputs to generate CVT-based aggregation boundaries for the building exposure model for seismic and tsunami risk assessment. The spatial population density (PD) in Lima at the block level (INEI, 2017) has been

combined with a "worst-case" scenario of tsunami inundation height (TI) obtained from a Mw 9.0 tsunami scenario. The distribution of the GMPE-based ground motion has not been used due to the reasons outlined in Sect. 3.2 (i.e., absence of site-response analyses). Hence, due their high spatial correlation, the visualisation of seismic-intensity-driven hot-spots would not be representative within a focus map. Both map layers have been linearly normalized and combined using the log-linear pooling expressed in Eq. (2) in order to assign a higher probability to the spatial locations where both indicators are relevant. Two sets of

weights that rank and combine the layers have been selected to perform a sensitivity analysis at this step. In both sets, tsunami intensities were ranked higher as population density due to their lower spatial correlation. The following weights were accepted for the construction of the two focus maps: set (1) PD = 30%, TI = 70% and set (2) PD = 40%, TI = 60%. The resulting focus map for the first set is shown in Figure 5. These models are available in Gomez-Zapata et al. (2021c).

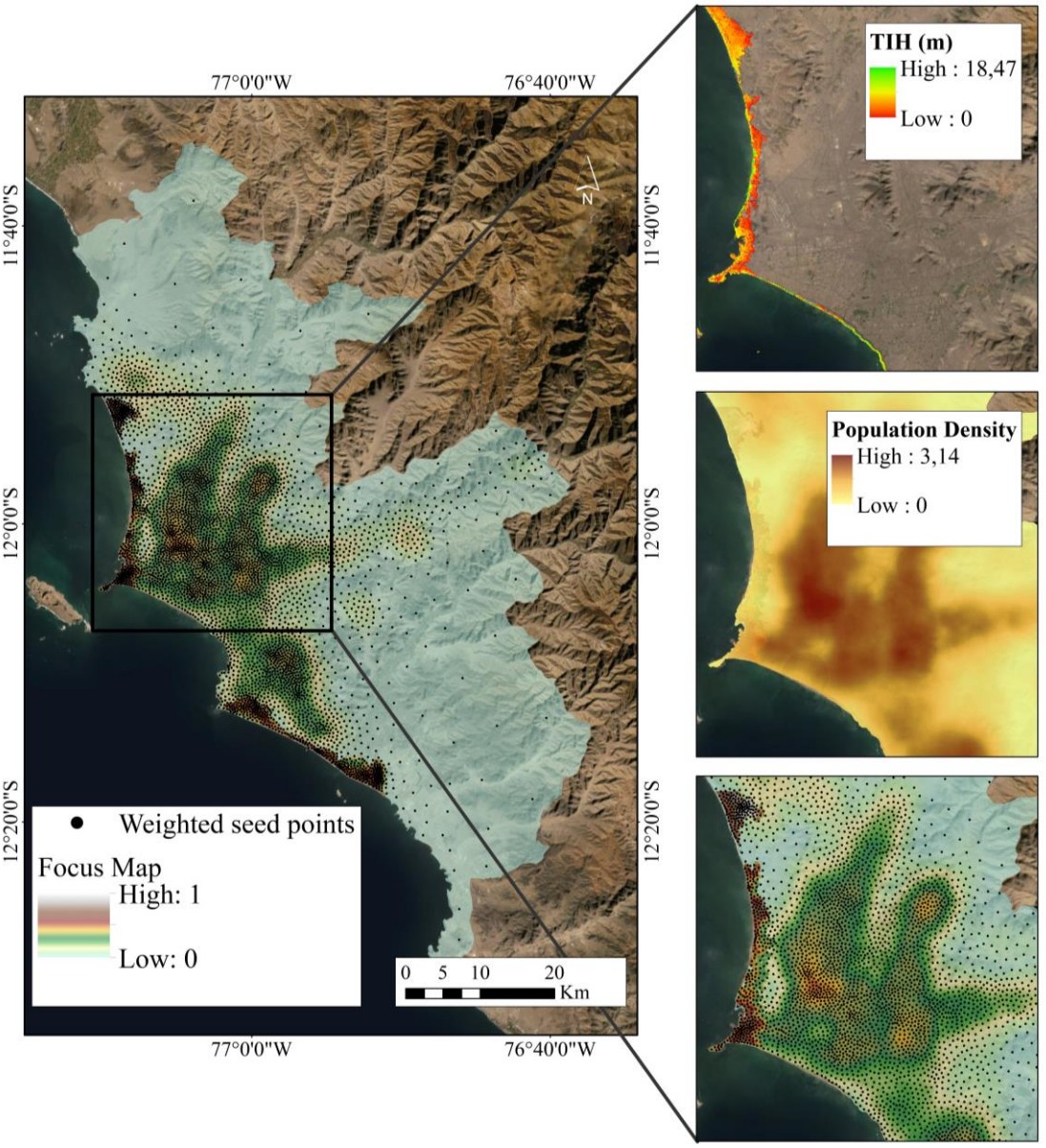

**Figure 5.** Example of the construction of focus map and CVT models for Lima. (a) 5,000 weighted seeding points sample a focus map through a Poisson point process. The normalised focus map is constructed from a log-linear pooling algorithm of the combined layers (population density (PD) and tsunami inundation height (TI) with a selection of 30% and 70% weights respectively). Map data: ©Google Earth 2021.

**3.4 Generation of CVT-based exposure aggregation boundaries.**

Three seeding sets have been generated by sampling the heterogeneous Poisson point processes defined by the two focus maps including 5,000, 10,000 and 50,000 initial points. We obtained six CVT aggregation entities for residential building exposure modelling by applying the Lloyd relaxation method as described in Pittore et al. (2020) and recalled in Sect. 2.3. As an example, the resultant CVT-based model obtained from the focus map from set (1) and the 5, 000 seeding points (model PD30_TI70_5,000) is depicted in Figure 5. The area jointly exposed to the earthquakes' ground motion and the largest tsunami footprint (Mw 9.0) is

highlighted in pink colour. The distribution of $Vs_{30}$ values in the study area within this CVT-based example is shown in Figure 7. Due to the contribution of the population density layer (PD), for every $Vs_{30}$ value at each location, there is a higher density of IM values that are computed where the exposed assets are expected to be concentrated rather than in the locations less densely populated, in contrast with what would occur using a regular grid.

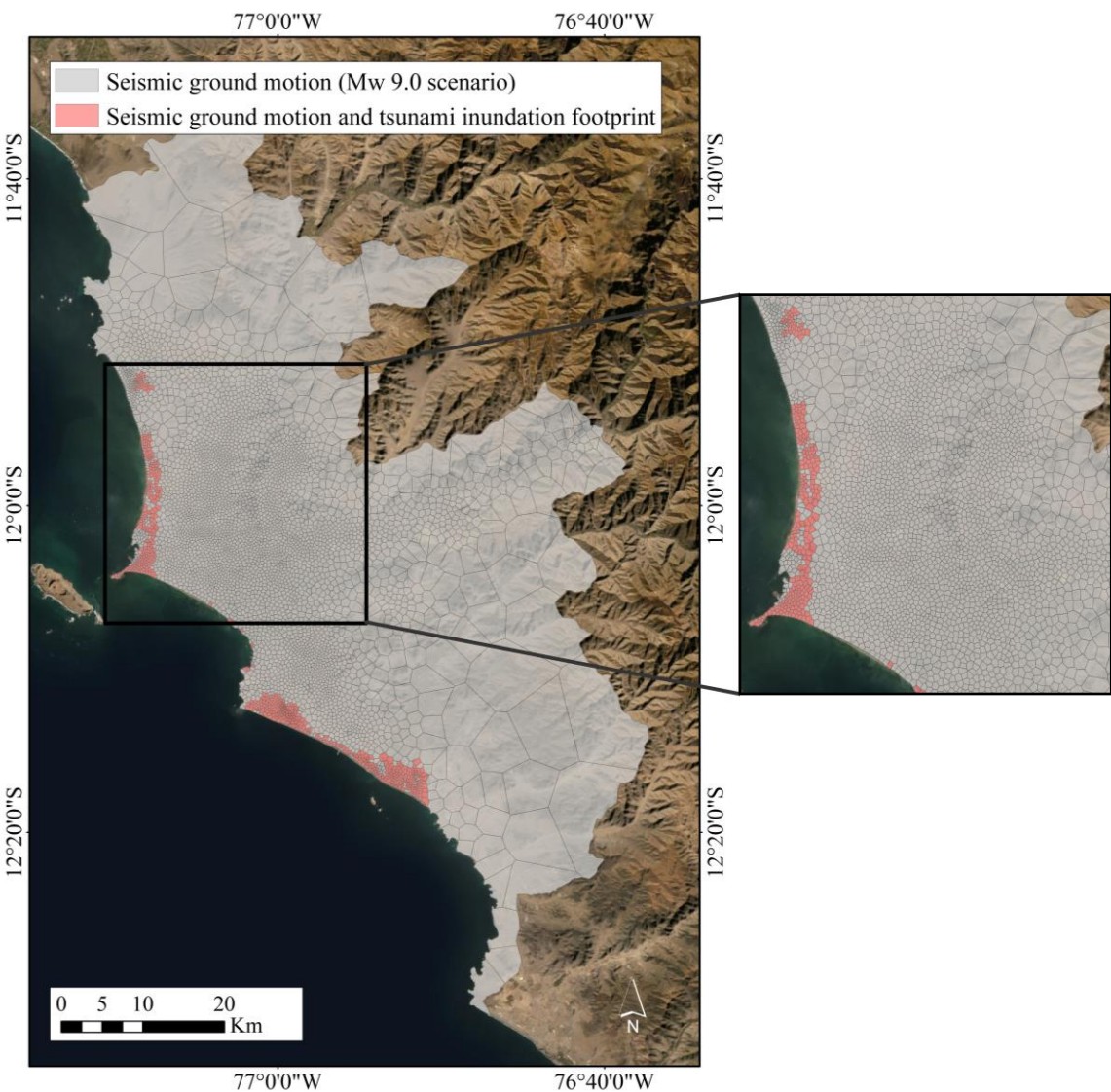

**Figure 6.** The resultant CVT geocells. The common exposed area to a Mw 9.0 earthquake and tsunami is coloured in pink whilst the area only exposed to seismic risk is coloured in grey. Map data: ©Google Earth 2021.

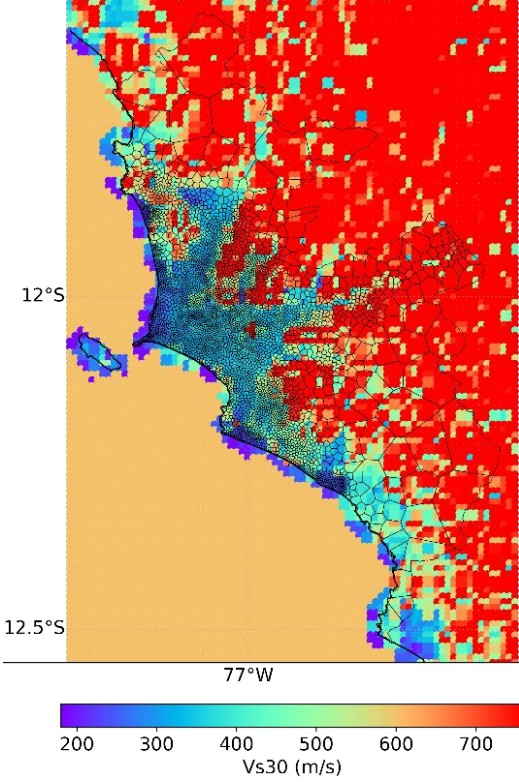


**Figure 7.** Spatial distribution of Vs$_{30}$ values in Lima/Callao as reported by Ceferino et al. (2018b) enclosed within the CVT-based model PD30_TI70_5,000.

### 3.5 Classification of the building stock into vulnerability classes and aggregation

The residential building stock of Metropolitan Lima (Peru) has been classified in terms of one scheme oriented towards seismic

vulnerability and two tsunami-related schemes with related building classes. They have been constructed following Sect. 2.4. The logical steps are depicted in the flowchart shown in Figure 8. The initial input is the official census dataset for Lima compiled by the Peruvian statistics institution (INEI, 2017) at the block level. It provides the number of buildings for each block and a few exposure attributes regarding the type of dwelling, floor, and façade predominant materials at the dwelling level. The mapping-scheme proposed through expert elicitation in the SARA project (GEM, 2014; Yepes-Estrada et al. 2017) for Peru has been used

to relate the census attributes with the proportions expected for 21 building classes. Subsequently, the dwelling fractions (per building unit) proposed in the same study have been used to obtain the building counts for every urban block. The building portfolio is therefore spatially distributed into every CVT-based model through a simple disaggregation procedure addressing their mutual intersections with the block-based model.

Two tsunami reference schemes are selected to classify the building stock of Metropolitan Lima, namely Suppasri et al.

(2013) and De Risi et al. (2017) to explore the epistemic uncertainty in their classification. While the first one addresses ten building classes in terms of predominant material and number of stories, the second only accounts for four classes based solely in terms of building material. Steel classes are not included since they have not been deemed representative in Lima (Yepes-Estrada et al. 2017). Thus, we retain seven and three classes, respectively. Considering SARA as the source scheme, the approach presented in Gomez-Zapata et al. (2021) was used to obtain the SARA - Suppasri et al. (2013) and SARA - De Risi et al. (2017) inter-scheme

compatibility matrices shown in Figure 9. Through their use, the building stock is represented in terms of the building classes of the target tsunami schemes. An example of how to calculate these matrices can be consulted in Gomez-Zapata et al., (2021c)

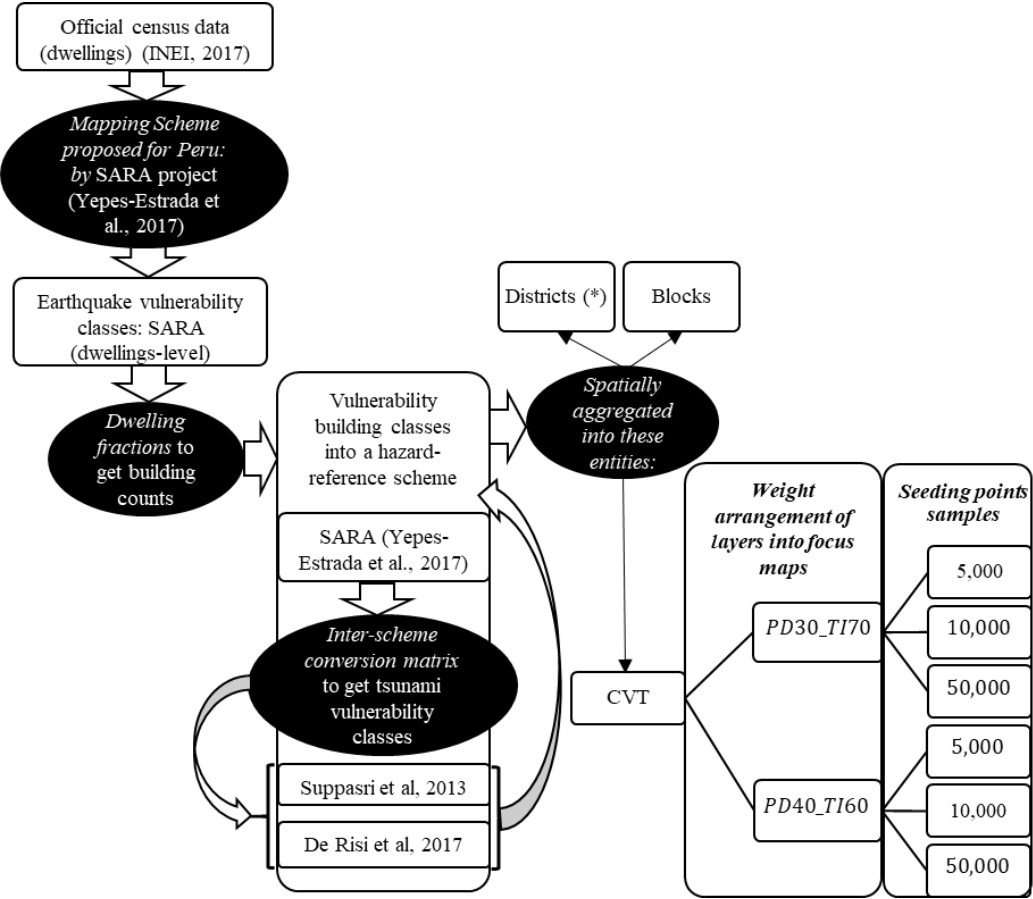

**Figure 8.** Flowchart outlining the process for constructing the building exposure model for Metropolitan Lima, including the condition tree used for the construction of CVT-based exposure models for the aggregation of earthquake and tsunami vulnerability building classes. (*District-based aggregation entities are only used for seismic risk to compare absolute loss values).

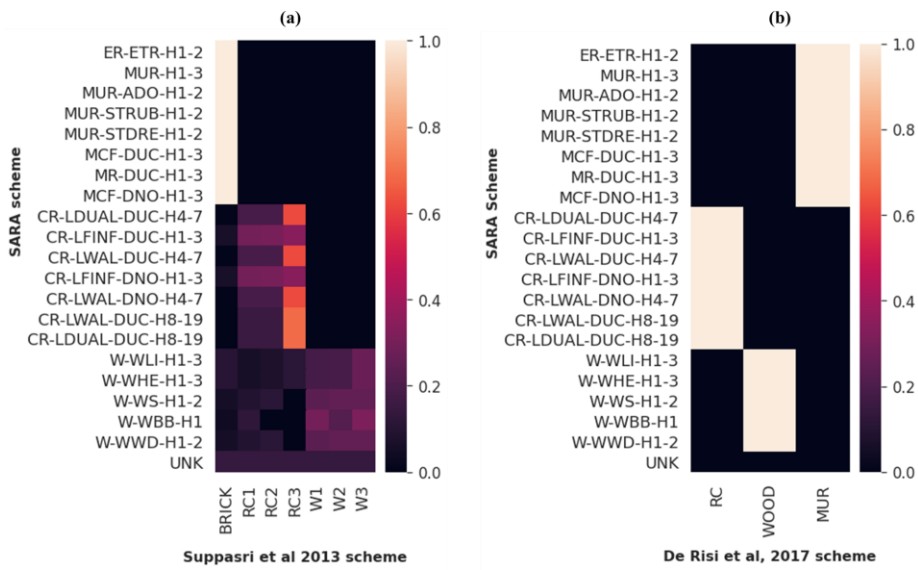

**Figure 9.** Inter-scheme compatibility matrices for Lima showing the compatibility level between the seismic-oriented reference scheme SARA and the tsunami-oriented target schemes: Left: Suppasri et al. (2013) and Right: De Risi et al. (2017).

Every building portfolio for the two considered hazards is aggregated upon the block-based aggregation entities: the six CVT-based and, for the seismic risk (using SARA), over the Peruvian third administrative level division (districts). The building class frequency distribution in the "La Punta" sector (Callao) is depicted in Figure 10a,b in terms of the seismic oriented- SARA scheme and in Figure 10c,d in terms of the two selected tsunami schemes. These models are available in Gomez-Zapata et al., (2021d).

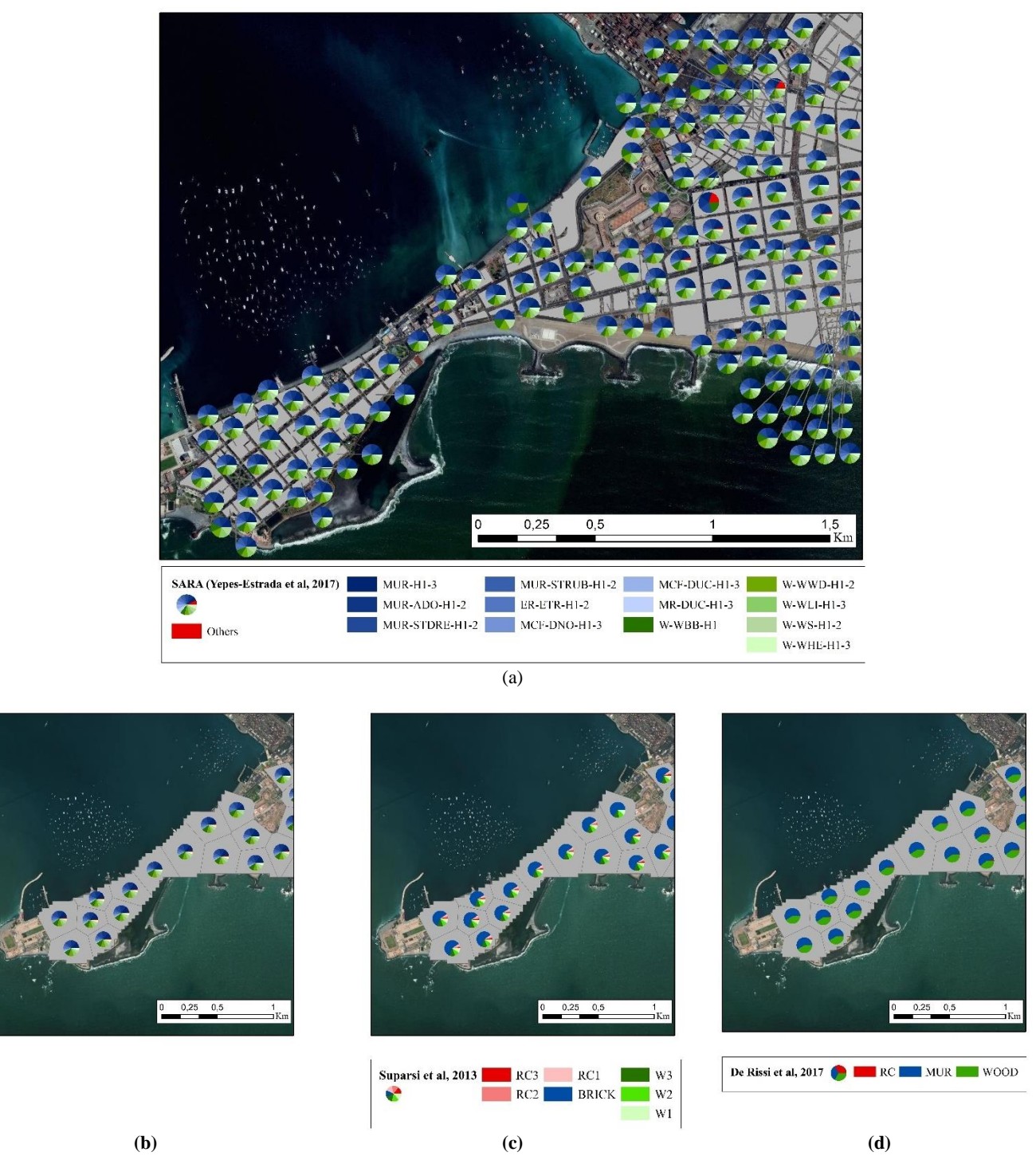

(a)

(b)        (c)        (d)

**Figure 10.** Example of the building class frequency distribution in "La Punta" (Callao) mapped using the seismic oriented- SARA scheme (Yepes-Estrada et al., 2017) (a) At the block level, (b) at the CVT based model PD30_TI70_5,000. The latter model is used to aggregate the tsunami vulnerability oriented building classes: (c) Suppasri et al. (2013) and (d) De Risi et al. (2017). Map data: ©Google Earth 2021.

## 3.6 Comparisons of aggregation areas for exposure modelling

As suggested by Petrone et al. (2020), due to the fundamentally different structural responses to both perils, the direct economic losses of the aggregated building portfolios for the six scenario earthquakes and the corresponding tsunamis have been separately estimated. The variability of the aggregation areas that form every residential building exposure model of the entire Lima/Callao is depicted in Figure 11a and listed in Table 2a. Conversely, if we narrow down the exposed area to the largest tsunami footprint (Mw 9.0), we see that the variability in the aggregation areas differs greatly (Figure 11b). The CVT-based models with higher resolution geo-cells (50,000) can reach very small areas when the focus map considers the weights PD = 30%, TI = 70%, whilst the block model can reach the largest area values. Moreover, the model PD30_TI70_50,000 provides a larger number of geo-cells and has a similar representation area with respect to the non-contiguous block-based model (see Table 2b). Furthermore, from Table 2 we can see that the computational effort (in terms of file size) required to construct the various exposure models is heavily dependent upon the resolution and, hence, the number of geocells.

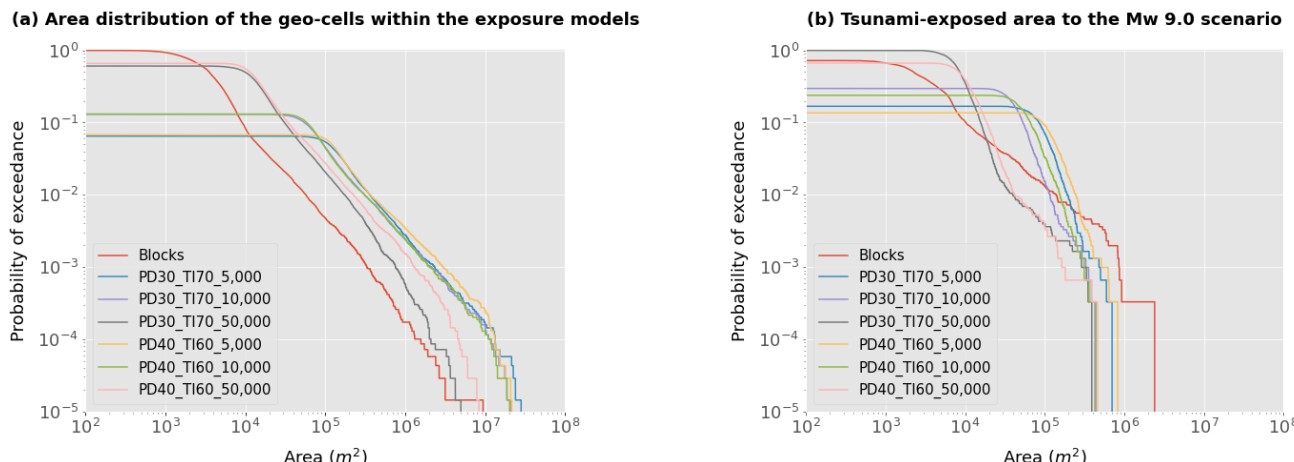

**Figure 11.** Variability in the area (in meters square) of the geocells that compose every aggregation area for exposure modelling, for (a) the entire urban area of Lima, and (b) for the area for which tsunami-induced loss values were obtained for the Mw 9.0 scenario. Seven models are evaluated: the official administrative block-based model and six CVT. The percentages assigned to the two focus maps' components (PD= Population density, and TI= tsunami inundation height) are written and are followed by the selected number of sampling seeding points.

**Table 2.** Variability of area (km²) and file-size (MB) across the exposure models proposed for (a) the entire urban area of Metropolitan Lima, and (b) for the area exposed to the tsunami from the Mw 9.0 scenario event. Only geocells with an urban land use are considered.

| Exposure model | **(a)** Configuration in the entire urban area of Metropolitan Lima | | | **(b)** Mapped as having suffered tsunami- induced loss from the Mw 9.0 scenario | | |
|---|---|---|---|---|---|---|
| | Number of geocells | ~Area mapped (km²) | Input size file (MB) | Number of geocells | ~Area mapped (km²) | Output size file (MB) |
| PD30_TI70_5,000 | 4,544 | 1,500.19 | 5.3 | 513 | 54.19 | 0.220 |
| PD40_TI60_5,000 | 4,722 | 1,695.82 | 6.0 | 416 | 57.45 | 0.227 |
| PD30_TI70_10,000 | 9,124 | 1,559.96 | 11 | 906 | 47.49 | 0.431 |
| PD40_TI60_10,000 | 9,182 | 1,554.14 | 10.5 | 728 | 51.49 | 0.302 |
| PD30_TI70_50,000 | 42,509 | 1,194.38 | 45.9 | 3,044 | 32.50 | 1.100 |
| PD40_TI60_50,000 | 46,217 | 1,537.82 | 53.7 | 2,034 | 28.42 | 1.010 |
| Block-based | 69,786 | 468.88 | 118.6 | 2,203 | 29.66 | 1.700 |

## 3.7 Results: scenario-based risk assessment

Tsunami and seismic risk assessments on classified residential building portfolios are carried out using the publicly available software DEUS (Brinckmann et al., 2021).

### 3.7.1 Seismic risk

Seismic losses for the entire study area are initially presented for a Mw 8.8 earthquake scenario so as to discuss the implications of the resolution of the exposure model in the economic loss estimates as well as on their associated mapping and visualisation. A comparison for the other five earthquake scenarios is provided in section 3.7.3 for the commonly exposed area to ground shaking and tsunami inundation. As described, the residential building stock of Lima is classified in terms of the SARA scheme and aggregated considering eight different geographical models (six CVT-based, one block, and one district-based model). Each

building class has an associated analytically derived fragility function provided in Villar-Vega et al. (2017) as well as their respective economical replacement cost reported in Yepes-Estrada et al. (2017). We have assumed loss ratios of 2%, 10%, 50%, and 100% as suggested by FEMA (2003) for each of the four damage states considered in the vulnerability model. Similar values have been recently proposed for seismic risk applications (e.g., Martins and Silva, 2020).

The seismic vulnerability analysis is performed at every geocell-centroid, where the buildings are aggregated. We consider

each IM value resulting from 1,000 realisations of spatially cross-correlated and uncorrelated ground motion fields. The resultant distributions for the Mw 8.8 scenario are displayed in Figure 12. Uncorrelated ground motion fields led to very homogeneous distributions, except at the district level. This finding is aligned with the recent study presented by Scheingraber and Käser (2020). Moreover, the latter confirms that if the dimension of the geocells in the exposure model is larger than a typical seismic ground motion correlation length (i.e., 20 km), an artificial bias in the ground motion correlation has to be expected as described in Stafford

(2012). We obtain larger median loss values from uncorrelated ground motions. We observe that for the considered scenario, the median loss values are insensitive to the aggregation of the exposure model at varying resolutions. This feature was already described in Bal et al. (2010) for a crustal earthquake damaging a building portfolio in Istanbul while neglecting the cross-correlation model. We thus confirm this finding while expanding it to when a ground motion cross-correlation model is considered.

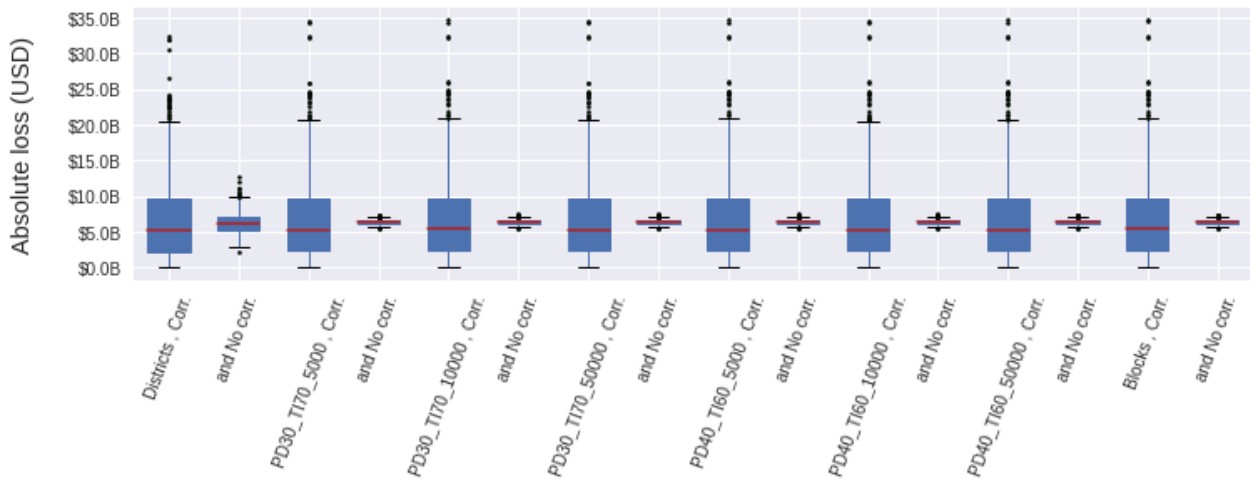

**Figure 12.** Computed loss distributions from a Mw 8.8 scenario for the residential building stock of Lima classified in terms of the SARA vulnerability classes aggregated into eight geographical entities. Two ground motion field conditions are analysed in every case, namely with the selected cross-correlation model (Corr.) and with uncorrelated ground motion fields (No Corr.).

The financial loss results that we have obtained are similar to the loss distribution estimated by Markhvida et al. (2017), investigated the possible losses of the residential building stock of Lima/Callao (aggregated into a regular grid (~ 1 km$^2$)) expected

from a similar Mw 8.8 scenario, who reported mean loss values of around 7 and maxima of around 35 billion USD. Although the authors employed a different GMPE from the one we adopted, the ground motion cross-correlation model, as well as the set of building classes and fragility functions, are the same as what we have implemented.

In Figure 13 the spatial distribution of losses for the considered Mw 8.8 scenario are shown for the "La Punta" sector for two selected ground motion realisations. In accordance with Vamvatsikos et al. (2010) and similarly to other studies entailing the

spatial comparison of risk estimates (e.g., Senouci et al., 2018), the loss values are normalized.

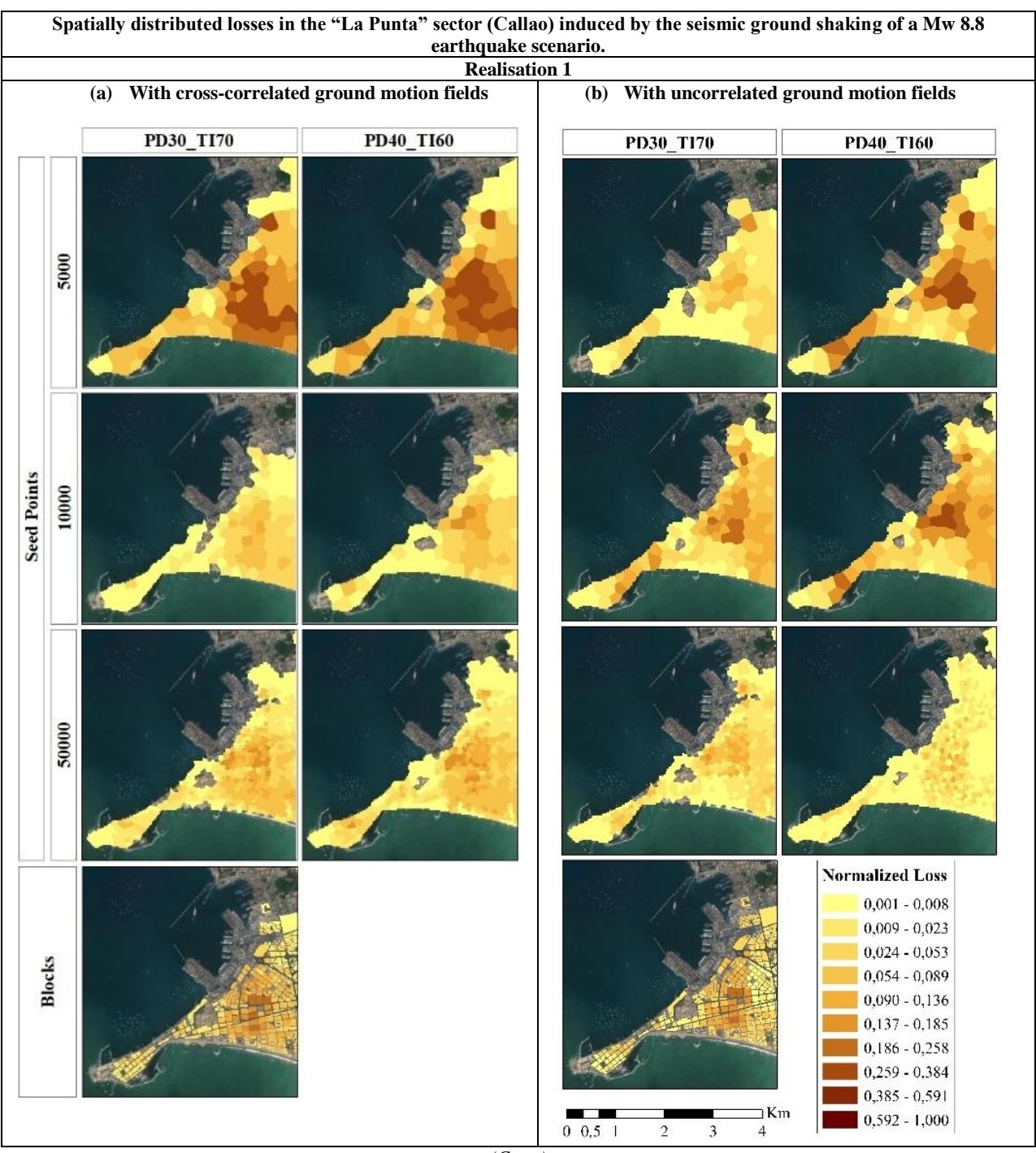

*(Cont.)*

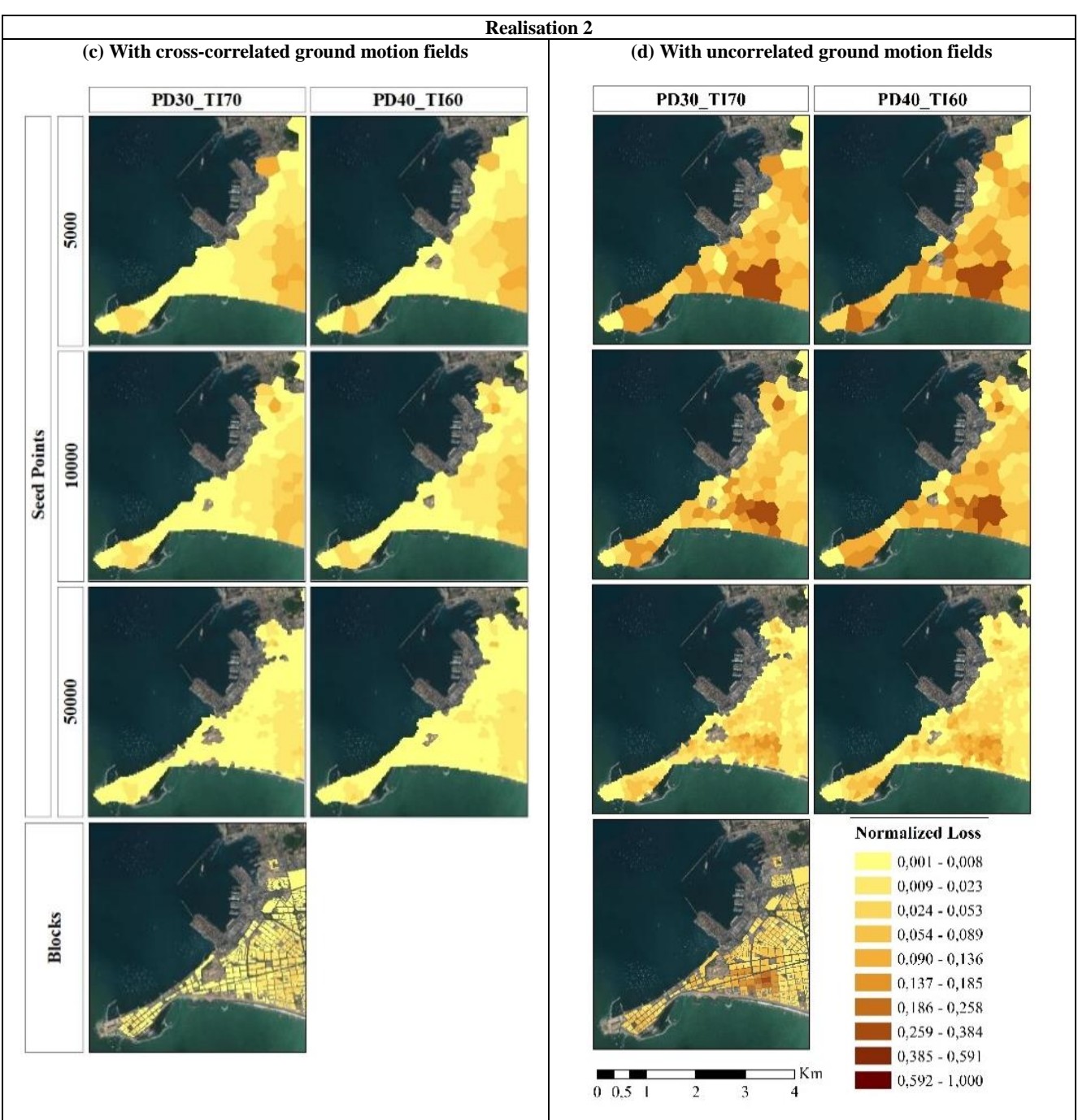

**Figure 13.** Spatially distributed losses in the "La Punta" sector (Callao) induced by seismic ground-shaking of a Mw 8.8 earthquake scenario. They are mapped over six aggregation areas of the building portfolio classified in terms of the SARA vulnerability classes. This is done for two randomly selected realisations with uncorrelated ground motion fields (b, d) and cross-correlated ground motion fields (a, c) using the Markhvida et al. (2018) model for PGA and spectral acceleration at periods 0.3 s and 1.0 s. Map data: ©Google Earth 2021.

Despite the remarkable differences between the area distributions of the models (Figure 11, Table 2), we do not observe significant differences in the absolute seismic-induced losses, which might be explained by the high special correlation of seismic ground motion and the resolution of the Vs$_{30}$ geo-dataset implemented. However, large differences arise when the normalised losses are mapped. It can be noted that for the same realisation, regardless of the use of correlated or uncorrelated ground motions, the seismically vulnerable areas are still identifiable, albeit with considerable differences. The use of cross-correlated ground

motion fields results in smoother mappings. However, the component which imposes the largest impact on the loss estimated from scenario earthquakes is the simulation of the seismic process, as remarked in other studies (e.g., Silva, 2016). This further highlights the importance of using quantile analysis in mapping seismic risk estimates for better visualisation and communication of the uncertainties in an inherently stochastic process (Geller, 2015).

### 3.7.2    Tsunami risk

To constrain the economical consequence model used in the tsunami risk assessment, the inter-scheme conversion matrices depicted in Figure 9 are used to obtain the replacement cost values per building class from the corresponding maximum scoring class in SARA. We have assumed loss ratios 5%, 15%, 45%, 65%, 85%, and 100% for each of the six damage states proposed by Suppasri et al. (2013) and similarly, but starting with 15%, for the five ones proposed in De Risi et al. (2017). A similar approach has been recently adopted by Antoncecchi et al. (2020). The impact of using more exhaustive approaches (e.g., Suppasri et al., 2019) is worth exploring, but out of the scope of this paper. Both tsunami-vulnerability schemes have been associated with a set of empirical fragility functions with tsunami inundation height in meters as the IM. They were derived from the same building damage dataset collected after the great 2011 Mw 9.1 Japan earthquake and tsunami with damage state definitions that implicitly accounted for the combined effect of both hazardous events. Thus, despite the extensive use of empirically derived fragility functions from that specific near-field event, care should be taken when using them, not only because they also account for the ground-shaking induced damage, but also because a submarine landslide could have contributed to the tsunami (Tappin et al., 2014). Nevertheless, there is a profound difference in the way the mean intensity values were obtained. Whilst in Suppasri et al. (2013) a linear least squares regression fitting was carried out, in De Risi et al. (2017), a multinomial logistic regression was performed for material-based classes. The latter found similar regression values to the case when an average simulated flow velocity of 1.84 ms$^{-1}$ for masonry and wooden buildings classes (predominant in Lima) is integrated into a hybrid fragility model. Making use of these two schemes, we have correspondingly estimated the tsunami-induced losses for the six scenarios and for the seven exposure models. Tsunami loss estimates normalised to the losses by the block-model are presented in Figure 11. Independent of the reference scheme, the two CVT models with the largest number of geocells (50,000) show the closest similarity to the block model (normalized ratio ~1). However, for all the CVT models, this ratio dramatically drops for scenarios with lower magnitudes (8.5, 8.6, and 8.7) which can probably be explained by a smaller tsunami footprint and lower IM spatial correlation.

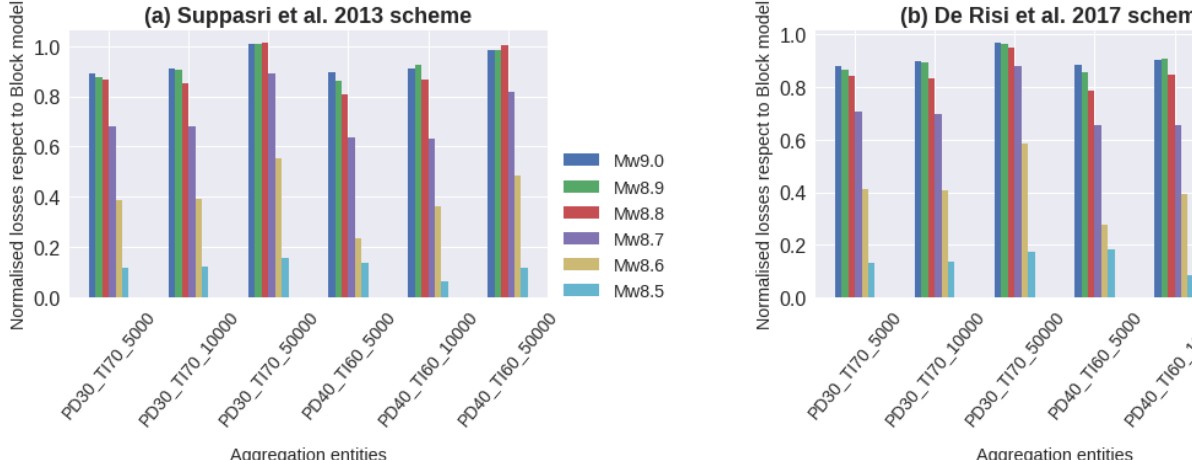

**Figure 14.** Losses induced by six tsunamis for the six CVT models normalised with respect to the ones at the block level. Tsunami vulnerability has been computed using the set of building classes proposed in (a) Suppasri et al. (2013) and (b) De Risi et al. (2017).

The absolute loss values expected after the six tsunami scenarios are reported in Figure 15 at the block level for the two vulnerability reference schemes. The fragility models of Suppasri et al. (2013) predict larger values with respect to the model proposed by De Risi et al. (2017) whose functional values were found within the range as if flow velocity was accounted for. These findings are in line with the observations of Park et al. (2017); and Song et al. (2017). These studies concluded that flow-depth models predict higher probabilities of complete damage for buildings than models that employed tsunami velocity in their derivation. Nevertheless, the aggregation of various building classes into less diversified schemes (e.g., only in terms of construction material in De Risi et al., 2017) might also have influenced the results due to the simplifications involved in the assigning of the financial consequence models. Crowley et al., (2005) described a similar effect for seismic risk applications.

We have computed the discrepancy in the tsunami loss estimations obtained for each CVT model with respect to the block-based model (Figure 16). This is minimised for the larger magnitudes and higher resolution models (50,000 geocells). This analysis shows that the Suppasri et al. (2013) fragility model leads to slightly larger differences (with respect to the block-based model) for the three lower magnitudes, whereas De Risi et al. (2017) shows larger differences for the three larger ones. These differences are minimised for the largest resolution model.

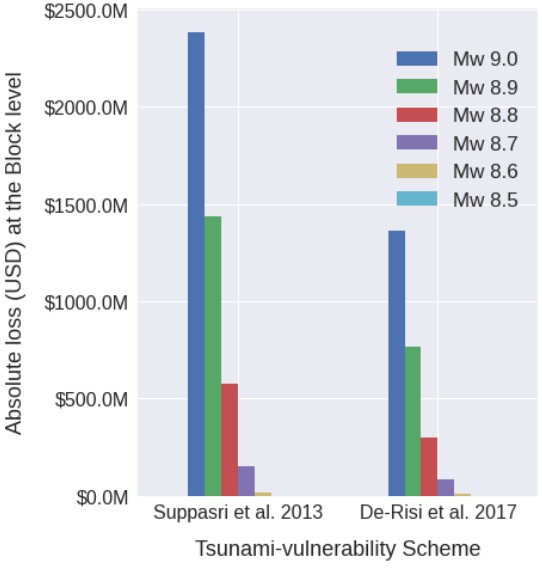

**Figure 15.** Absolute losses (USD) for six tsunami scenarios for the residential building portfolio of Lima classified in terms of two reference schemes and aggregated at the block-based model.

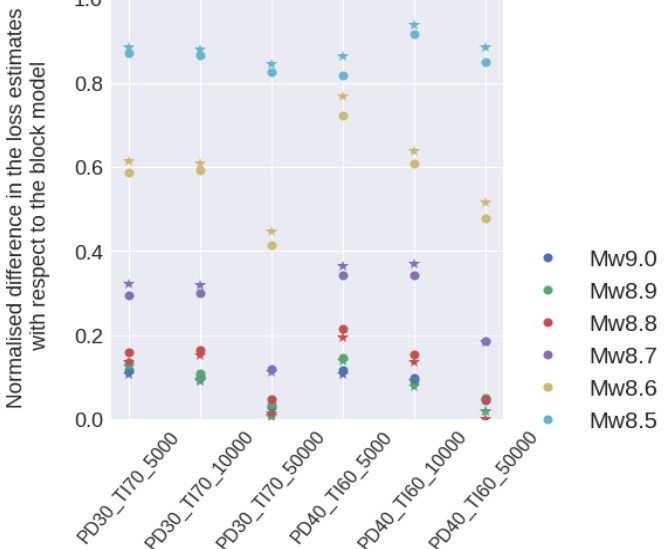

**Figure 16.** Discrepancy between the tsunami-induced losses between each CVT-based model and the block model for the six scenarios. The values obtained from the Suppasri et al. (2013) and De Risi et al. (2017) schemes are denoted by stars and circles respectively.

Estimates at the block level are around 3 times less efficient (i.e., much larger file sizes, see Table 2) compared to the CVT-models with 50,000 geocells for which almost identical results are obtained (Figure 14), and around 23 times greater than CVT-models with 5,000 geocells for which overestimations by less than around 20% are expected for the three largest magnitudes (Figure 16). Aggregation entities with lower resolutions lead to overestimations in the tsunami-induced losses. Similar findings were reported in Figueiredo and Martina (2016) in a flood risk analysis.

Tsunami loss outcomes for the Mw. 8.8 scenario are mapped and discussed hereafter for the residential building stock in "La Punta" (Figure 17) and Chorrillos district (Figure 18). Only geocells with loss values larger than zero are colour mapped. Due to the normalised metric used, no significant differences in the tsunami vulnerability mapping induced by the independent building classification schemas are noticeable. The CVT models at the coarser resolutions (first two rows in every figure) show the largest values, and hence overestimations compared to the block level and other finer CVT- models. Overestimation of losses decreases with the increase in resolution. Due to the adjacency and compactness of the highest resolution CTV model (fourth row), for "La Punta" we identify at least four zones with a comparatively higher tsunami vulnerability.

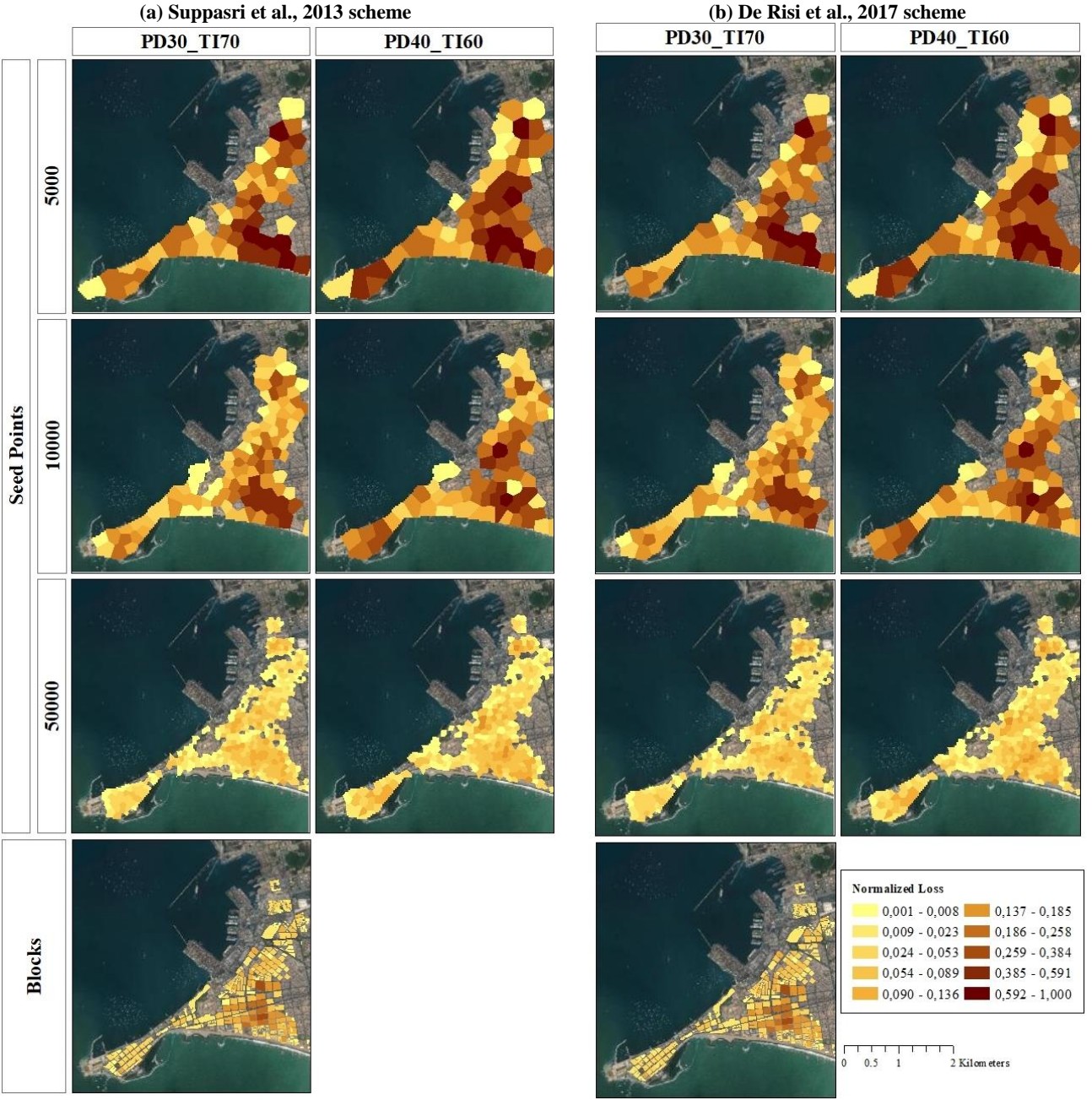

**Figure 17.** Spatial distribution of tsunami-induced normalized losses (Mw 8.8 scenario) for the La Punta sector (Callao district) using two tsunami reference schemes (a) Suppasri et al., 2013 and (b). De Risi et al., 2017. Map data: ©Google Earth 2021.

Considering Figure 18, it can be noted that the overall mapped area is increasingly reduced as the resolution of the CVT models increases. This is due to the lack of residential buildings within the three large parcels, namely *"Country Club de Villa"*, *"Reserva*

*Laguna de Villa"*, and *"Refugio de Vida Salvaje Pantanos de Villa"* that occupy most of the exposed area in the block-based model. These zones represent the largest area values in Figure 11b. This model assigns the largest loss values in the Chorrillos district to these three large blocks due to the assumption of using a single tsunami intensity as being representative of the entire enclosed area. Therefore, if the block polygons are too coarse compared to the hazard footprint and IM spatial correlation, biases in the loss assessment are expected. This is highlighting the importance of hazard-driven entities for exposure spatial aggregation.

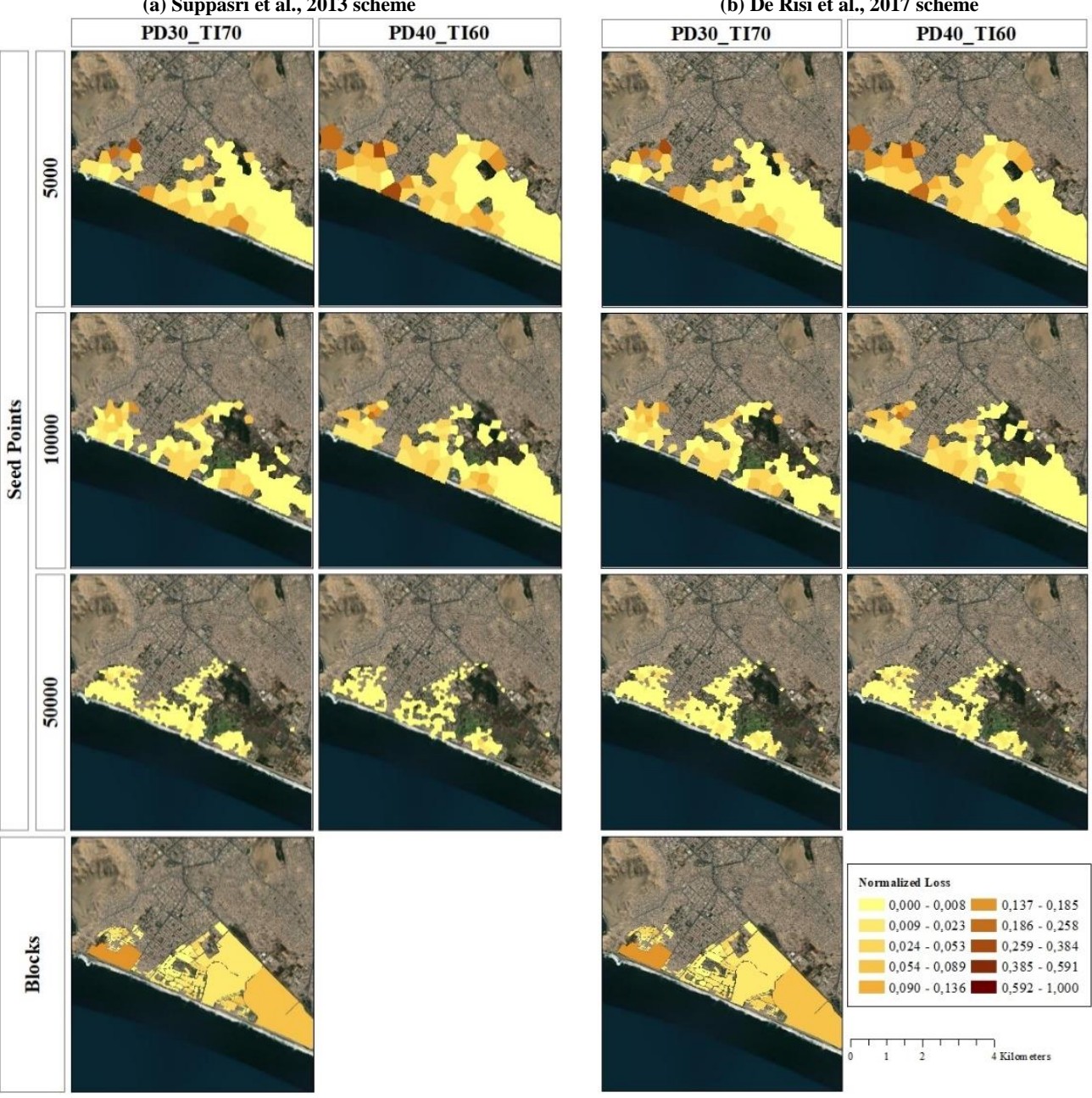

**Figure 18.** Spatial distribution of tsunami-induced normalized losses (Mw 8.8 scenario) for the Chorrillos district (Lima) using two tsunami reference schemes (a) Suppasri et al., 2013 and (b). De Risi et al., 2017. Map data: ©Google Earth 2021.

### *3.7.3*     **Comparison between earthquake and tsunami scenario-based induced losses**

In Figure 19 we compare the absolute losses induced by each hazard scenario onto the building portfolio exposed to both perils (e.g., Mw 9.0 in Figure 5-b). The CVT-based PD30_TI70_5,000 is used to represent the earthquake-induced losses. The latter was compiled for the cases with and without the ground motion cross-correlation model, each sampled with 1,000 realisations. Due to the lack of stochastic realisations in the tsunami case, the respective loss distributions were constructed for both reference schemes with the seven values obtained from the various aggregation entities (6 CVT- and 1 block-based models). Even though the

distributions for seismic and tsunami losses have been obtained independently, the median values are nevertheless illustrative for comparative purposes.

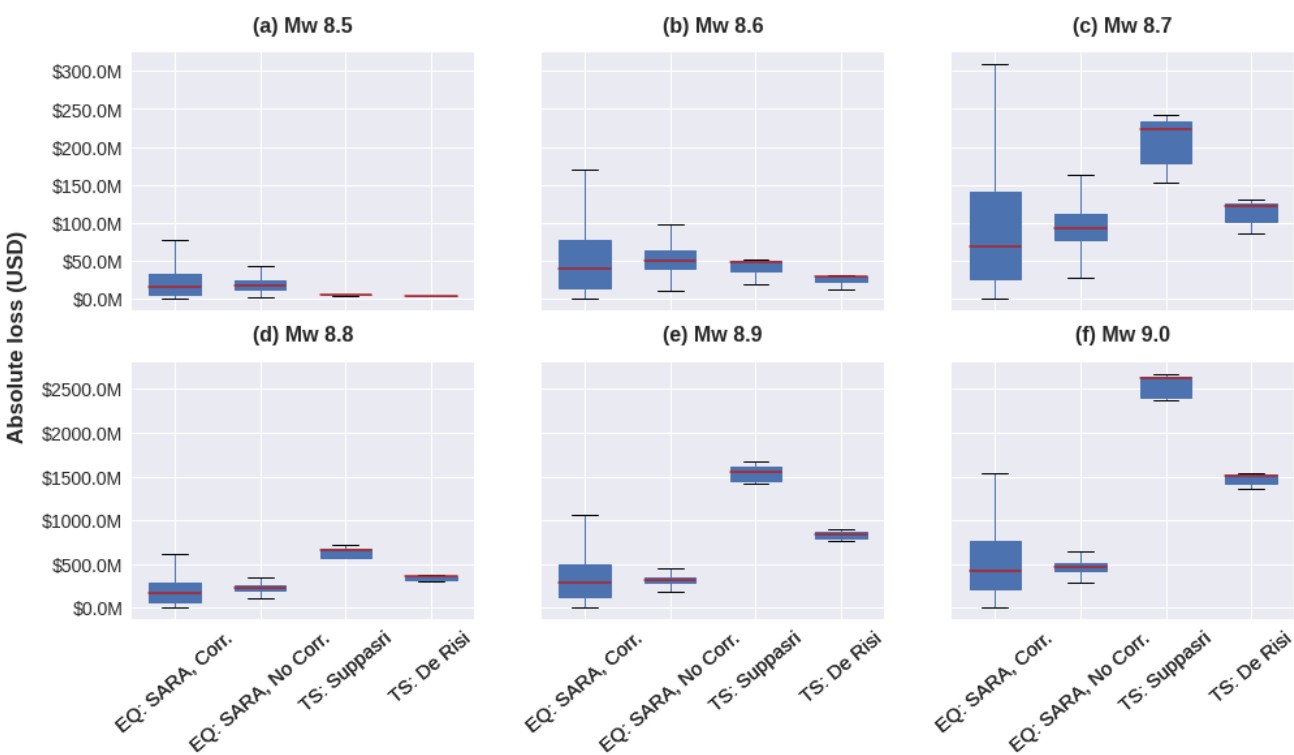

**Figure 19.** Comparison of the independent earthquake-induced losses (EQ) for two ground motion field conditions (using the 1,000 GMF for

each case: with a spatially cross-correlation model (Corr.) and spatially uncorrelated (No Corr.); and the tsunami-induced losses (TS) under two tsunami reference schemes for six magnitude scenarios over every common area exposed to both perils.

We observe that in our estimations for the commonly exposed area to both perils in Lima, the earthquake event dominates the

median losses at lower magnitudes (Mw 8.5, 8.6) whilst the tsunami prevails in the larger ones. The tsunami-induced median losses start to be larger than the earthquake-related ones for the Mw 8.7 scenario, although the latter still present high volatility in the extreme values due to the variability in the seismic realisations. From Mw 8.8 on, the tsunami-induced losses are always larger regardless of the tsunami reference scheme implemented. Our findings regarding the role of the earthquake magnitude in the disaggregation of financial loss estimates for every hazard scenario are in line with the results of Goda and De Risi (2018) obtained

for a coastal town in Japan that was jointly exposed to two decoupled earthquakes and tsunamis risk scenarios (Mw 8.0, 9.0).

## 4    Discussion

The derivation of CVT-based aggregation entities for building exposure modelling is subject to epistemic uncertainties, namely the selection of weights for the pooling of geospatial layers into the focus map and the selection of seeding points that provide the initial seeding set and control the overall number of geocells. Through the application of a condition tree, we have selected different sets of these two components to investigate the impact of customised CVT-based geographical entities to aggregate building portfolios as well as through the reconnaissance of thematic uncertainties in the loss mapping and visualisation. Voronoi regions inherently fulfil spatial properties such as compactness and contiguity which are useful to identify areas with comparatively homogeneous physical vulnerabilities.

CVT-based aggregation entities for building exposure modelling can be further customized. For instance, the underlying focus map can be modified in order to integrate other components such as seismic microzonations with higher resolutions than the one we have employed in this work; the spatial presence of certain building taxonomic attributes that may drive the physical vulnerability towards a given hazard (e.g., soft-storey in seismic vulnerability and openings / building foundation in tsunami vulnerability (Alam et al., 2018)); high-resolution DEM (digital elevation model). However, caution should be taken not to double count their contribution if the hazard simulations have already been performed using these input data (e.g., DEM in landslide susceptibility and tsunami inundation) as well as a wise selection of their respective weights in the focus map construction. Furthermore, CVT model generation would benefit from further improvements such as outlining an iterative approach that can seek a minimum geocell size from a convergence criterion imposed by spatially correlated hazard-IM-lengths.

Several of the limitations in this study could be addressed in further studies. For instance, it is worth conducting sensitivity analyses that address the differential impact of the selection of other GMPE(s), as well as their combination in logic trees (Scherbaum et al., 2005). In addition, it is relevant to include the calculation of an exhaustive set of stochastic tsunami flood scenarios (with respect to the considered magnitudes) for the evaluation of losses. Likewise, having a higher resolution digital surface model with spatially distributed roughness values is likely to allow the generation of more accurate results. Physics-based tsunami fragility functions based on intensities more relevant to the building failure mechanisms such as momentum flux (e.g., Macabuag et al., 2016; Attary et al., 2017) would benefit future risk simulations. However, this is subject to their actual availability for typical Peruvian building classes. More comprehensive approaches to adapt such as "foreign" empirical fragility models (e.g., Suppasri et al., 2019; Paez-Ramirez et al., 2020), as well as the need of future development of analytical functions for the South American context (e.g., Medina et al., 2019) would benefit future risk assessment studies for Lima. Another area that would benefit from future research is the differential selection of loss ratios with dependencies on the building classes, as for instance recently investigated by Kalakonas et al. (2020) for seismic risk applications. This might be also relevant for tsunami-induced losses that are strongly influenced by the presence and cost of non-structural building elements. Accordingly, more refined financial tsunami consequence models such as the one proposed by Suppasri et al. (2019) and/or Triantafyllou et al. (2019) are worth exploring when detailed information about prices and built-up areas at the individual building level are available for the study area. In the presented example case, we make use of the concept of inter-scheme conversion matrices to further prove their usefulness to derive exposure models (i.e. spatial distribution of building classes and replacement costs). This is novel because, if we can know these characteristics for a single exposure scheme (e.g. seismic vulnerability oriented), we could get the same descriptors for another vulnerability scheme (e.g. tsunami). This procedure ensures the comparability across the different schemes and this compatibility had not been considered so far in the related scientific literature for multi-hazard exposure modelling. This aspect also outlines that various exposure models existing in the literature can actually be complemented and compared in a probabilistic manner. On the one hand, the latter ensures that the exposed residential assets classified under various schemes have approximate replacement costs, and thus, the hazard-dependent risk estimates can be comparable with each other. On the other hand, caution should be taken

when interpreting the presented results. Neither the damages induced by debris impacts nor scour, relevant for a clearer tsunami vulnerability assessment (Charvet et al., 2015), are included in our modelling. Moreover, it is worth mentioning that larger indirect losses can be expected from buildings with other occupancies (e.g., Chen et al., 2018) that we have not considered herein.

CVT-based models can be beneficial to define efficient, multi-hazard aggregation entities for earthquake and tsunami risk assessment, not only in Lima, but also in other coastal cities exposed to similar hazards. Furthermore, it is worth investigating the usefulness of mapping cumulative damage and losses in hazard sequences, i.e., when a first hazardous event modifies the fragility of buildings that are then affected by a successive event, e.g., an earthquake affecting and area that is then struck by a tsunami.

## 5    Conclusions

This work has introduced a novel contribution to derive spatial aggregation entities with variable resolution for large-scale building portfolios for physical risk assessment applications. To this aim, we have presented a workflow to find an adequate resolution of the exposure model where it really matters, i.e., in areas where buildings are densely distributed and/or hazard intensities vary over short distances. This contrasts with the current state of the art related to building exposure modelling (aggregation) that uses regular grids or purely administrative boundaries for exposure aggregation.

In the context of earthquake and tsunami risk, we take advantage of the focus map concept to integrate spatially correlated hazard intensity measures (IMs) with exposure proxies (i.e., population density) in order to spatially identify hot-spot areas where higher values from both spatial distributions are expected. These resultant focus maps can then be sampled by a heterogeneous Poisson point process, as proposed by Pittore et al. (2020) in order to generate variable-resolution aggregation entities in the form of Central Voronoi Tessellations (CVT). Each CVT geocell becomes a minimum resolution of risk computational analysis, handling the inputs (i.e., hazard intensities and exposure model) and output elements (i.e., damage and loss estimates).

Variable-resolution CVT-based exposure models proposed in this work have proved their efficiency in integrating large-area building portfolios for combined earthquake and tsunami loss estimations. Several advantages over conventional models based on administrative aggregation entities are:

- CVT-based models provide alternative an approach to aggregate an extensive building portfolio constructed from ancillary data (i.e., population) in the case when existing administrative aggregation areas are not suitable (either not publicly available, or too coarse in resolution) for a certain area of interest, as well as to perform scenario-based risk assessments for various hazards.

- We have observed that CVT-based models correct some bias in the spatial aggregation of buildings due to the smaller, more compact areas in high-resolution CVT geocells with respect to a coarser block-based cell. This correction is further propagated to the loss estimates due to the higher density of IM values employed by the respective fragility functions during the loss assessment. This is especially observed in areas of the largest concentration of exposed assets located within the hazard footprint area and where local spatial variations of the IM are expected, leading to more accurate estimates.

- They are computationally more efficient than the block-based models in earthquake and tsunami vulnerability assessments. This is advantageous when thousands of stochastic realisations of hazard scenarios are calculated over the aggregation boundaries that are used to model building portfolios.

- They have shown to be beneficial for mapping loss estimates in continuous space with adjacent and compact geocells. These features allow the spatial identification of zones with similar vulnerability to the hazards considered and within the area of interest. They contribute to a more intuitive visualisation and interpretation of the loss mapping and hence contribute to raising awareness about epistemic and thematic uncertainties in the loss mapping.

For the portfolio exposed to both perils in Lima, we have found that the expected median loss values induced by seismic ground-shaking are insensitive to the representation of the exposure model over varying resolutions. Thus, we confirm the findings of Bal et al. (2010) and expand them to the case when cross-correlated ground motion fields are considered. However, this contrasts with the tsunami loss results, whose differences with respect to a high-resolution model (i.e., block-based) decrease as the resolution of the CVT geocells increases. Similarly, these differences are remarkably minimised for incrementally correlated tsunami-intensities from the large magnitude tsunami scenarios (i.e., Mw 8.8, 8.9, 9.0). According to our observations, the adopted tsunami fragility model based solely on flow-depth as the IM and linear square fitting (Suppasri et al., 2013) predicts much larger tsunami-induced losses on the residential buildings portfolios in Lima than the model of De Risi et al. (2017), which was derived through multinomial logistic regression and with similar values as if the flow velocity was accounted for. For the residential building portfolio exposed to both perils, we have found that the earthquake scenarios dominate the losses at lower magnitudes (Mw 8.5, 8.6) whilst the contribution of the tsunami is dominant for larger magnitude events.

Bearing in mind the scope of this study, but also the limitations presented in the discussion section, we are not claiming that the economic losses we have obtained for the residential building stock of Lima are exhaustive. Instead, through the adoption of the condition tree, we have drawn a branched methodological workflow to explore the differential impact of the exposure aggregation models, and the selection of building schemes on the epistemic and thematic uncertainties that are embedded in scenario-based risk applications. As described by Beven et al. (2018), condition trees facilitate the communication of the meaning of the resulting uncertainties while providing a clear audit trail for their analysis that can be reviewed and evaluated by others (e.g., local experts and stakeholders) at a later date. This study also highlights the relevance of hazard-based aggregation entities for exposure modelling, risk computations, and loss mapping. Thus, the continuous understanding of those uncertainty sources will contribute to enhancing future risk communications, mitigation, and disaster management activities by local decision-makers.

*Code and data availability.* The codes and data models used in this paper have been made available in open repositories (Brinckmann et al., 2021; Gomez-Zapata et al., 2021a, 2021b, 2021e, 2021f; Harig and Rakowsky, 2021).

*Competing interests.* The authors declare that they have no conflict of interest. The funders had no role in the design of the study; in the collection, analyses, or interpretation of data; in the writing of the manuscript; or in the decision to publish the results.

*Funding.* The authors disclose receipt of the financial support for the research and publication of this article from the RIESGOS project (Multi-risk analysis and information system components for the Andes region), funded by the German Federal Ministry of Education and Research (BMBF) Grant No. 03G0876, as part of the funding programme CLIENT II – International Partnerships for Sustainable Innovations.

*Acknowledgements.* The authors want to express their gratitude to Glendy Linares, Waldor Arevalo, and Walter Tapia from the Peruvian Office of National Security and Defence (Ministry of Housing, Construction and Sanitation) for providing the INEI, (2017) census geo-dataset. Thanks to Catalina Yepes (GEM) for providing the SARA exposure model and mapping schemes for Lima. Thanks to Kim Knauer (EOMAP) for a unified topography and bathymetry data set for the study region. We thank Sandra Santa-Cruz and Nicola Tarque (PUCP), Miguel Estrada, Diana Calderón, Fernando Lázarez, (CISMID), Luis Ceferino, Mary Chris Suarez (YANAPAY), Alireza Mahdavi (AWI), and Omar Campos (DHN) for the fruitful discussions regarding the seismic and tsunami hazard and risk in Lima during some of the authors' visits to the city. Likewise, we thank Tiziana Rossetto, Dina D'Ayala, Ingrid Charvet, Carmine Galasso, and Juan Palomino (UCL) and Pierre Gehl (BRGM) for the feedback during the 2019 Multi-hazard EPICentre encounter in London. Thanks also goes to Heidi Kreibich (GFZ) for the invitation to the 2020-AOGS-EGU NtHazards virtual Seminar, during which we were inspired to continue this study by Anawat Suppasri (Tohoku University). Special thanks to Elisabeth Schöpfer (DLR), Cecilia Nievas, Graeme Weatherill, Henning Lilienkamp, Matthias Rüster, and Jörn Lauterjung (GFZ) for their

valuable advice during the elaboration of this study. We thank Dr. Mario Salgado-Gálvez and three other anonymous reviewers for their valuable feedback. We would like to thank Kevin Fleming for the careful proofreading.

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
