# Peer review of "Variable-resolution building exposure modelling for earthquake and tsunami scenario-based risk assessment. An application case in Lima, Peru"

_Natural Hazards and Earth System Sciences, 2021_

## Author Comment (AC1)

**Responses to Dr Mario Salgado (Reviewer 1)**

Thank you for your helpful review. Please find our answers to each of your comments below.

**1. General comments.**

**1.1.**>> "The topic of the paper fits well under the scope of the Journal and in my opinion, it should be accepted after addressing some minor comments that I include below"

Thank you. We really appreciate it.

**1.2.**>> "The manuscript is well organized and written (although a minor final review of English is suggested, there are a couple of typos and sentences that are not easy to read) and a careful review of recent references has been made".

Thank you for the nice comment about the structure of the paper. Following your advice, we have accordingly asked a native English speaker to provide us a strict language review. The new version has been significantly been improved in that regard.

**2. Specific comments**

>> "Although the focus of the manuscript is mostly on the variable resolution level of the exposure databases, authors include too loss analyses for different earthquake (and tsunami) scenarios to assess the sensitivity of the different aggregation levels in the results".

Indeed, the main focus of the paper is the construction of variable resolution building exposure models and to test their impact and benefits on risk assessment. Hence, we do not perform, *on purpose,* sensitivity analyses for different controlling parameters of the earthquake and tsunami scenarios to keep less degrees of freedom upon the main goal of the paper: exploring the uncertainties in the losses carried by the resolution of the exposure model.

2.1.>> "These loss analyses make some assumptions which consequences are not negligible and have been studied recently (even in some of the documents cited in the manuscript). For instance, for earthquakes with the characteristics used in the case study, for which each rupture has zones below Lima, assuming no cumulative damage (ground shaking and then tsunami wave) or not assessing the quasi-simultaneous occurrence of the losses can have consequences in the obtained results, mostly for the tsunami case."

In order to solely study the impact of the exposure resolution for the two-hazards, we make some assumptions that are clearly described since the beginning. Such as "We present decoupled earthquake and tsunami scenario-based risk estimates for the residential building stock of Lima (Peru)". We cite the recent study we are based on for that assumption (Petrone et al., 2020). However, in order to raise awareness of our assumptions, we have accordingly cited some recent studies that have explored the effects of cumulative damage (see end of the Introduction section).

We then totally agree with you: addressing cumulative damage is fundamental. The new manuscript version is better stressing this issue in the Discussion. Addressing this important but complex issue was beyond the scope of this paper which is already quite long. We have been working in another complementary publication regarding a novel methodology for multi-hazard risk. This has been already presented in the EGU-2020 conference (Gomez-Zapata et al., 2020). We expect to submit the associated actual journal paper with a rigorous methodology very soon. This explains the very last sentence presented in the Discussion section: *"Furthermore, it is worth investigating the usefulness of in mapping cumulative damage and losses in hazard sequences i.e. when a first hazardous event modify the fragility of buildings that are then affected by a successive event."*.

2.2.>> "The EQ footprint was generated using only 1 GMPE which is known to be a highly sensitive component in the risk results. A discussion about how capturing the epistemic uncertainty (by any of the traditional methods typically used in PSHA) may (or not) affect the proposed"

We totally agree. This selection is crucial. This is a special research topic of our team. However, the influence of the choice of GMPE, and other epistemic uncertainties, on the final loss estimates is not within the scope of this study. As formerly stated, we reduced on purpose the degree of freedom to analyse the impacts of the selection of the exposure models. The selected GMPE predicts the log median lnSa and standard deviation  $\sigma$  of the spectral acceleration Sa(T) at periods T  $\in$  [ 0.01~PGA, 0.3s, 1.0s]. We explore the impact of using spatially uncorrelated or cross-correlated ground motion fields, which is nevertheless a step forward capturing epistemic uncertainties related to ground-motion modelling.

In the revised version we will added a note for clarity in this regard as well as the importance and uncertainties linked to this assumption. We cite a study that advised the use of logic trees for capturing the uncertainty in the GMPE selection. (e.g. Scherbaum et al., 2005). We point out once again the reference of Weatherill et al., (2015) in this section. This is because of its clarity when it points out: *"the choice of GMPE may influence upon the results depending on whether the inter- and intra-event components are homo- or hetero-skedastic, or due to the manner in which soil nonlinearity is accounted for in the functional form"*. Please note that in the latter reference the GMPE selection was neither within its scope.

2.3.>> "The consideration of site-effects was performed using by combining two models with different resolution level (i.e. the city's microzonation and the Vs30 values when needed), made available in another study for Lima. However, being this a parameter that defines somehow the weights in the proposed aggregation scheme, a discussion of the possible impact of merging two datasets with different resolution to account for the soil response should be included in the manuscript".

Indeed we have used the shear wave velocity in the uppermost 30 meters depth  $(Vs_{30})$  as reported in Ceferino et al. (2018b). The resulting resolution of that dataset is 30 arcseconds (~ 1km). We agree with the reviewer that that resolution might be coarse and a short note has been accordingly added in the Discussion section to raise awareness in this regard.

On the other hand, the weights used to derive the aggregation schemes are assigned to the customised focus maps (see section 3.3). Please note that only the population density and tsunami inundation are used to define the two types of focus maps presented. The spatial distribution of Vs30 values was never used to derive them. Perhaps the presentation of Figure 5 has caused that confusion. This figure is only displaying one of the resultant CVT-modes on top of the spatial distribution of Vs30 values in Lima/Callao, but is not an input to their derivation.

2.4.>>"The nonlinearity of the soil response is assumed as negligible. However, the microzonation for Lima identifies zones with soils that typically have large nonlinear effects, particularly when subjected to large EQ intensities as the ones expected for events with Mw 8.5-9.0 (see zones III and IV). This aspect should be revised and discussed with more detail by the authors, instead of only pointing out to a reference which at the same time contradicts the findings and statements of others used".

Please note that that the sentence the reviewer refers is included in the section "Construction of the focus maps" as a manner to justify why we did not include the expected ground motion to derive the focus maps. Nothing similar was included in Section 3.2 that refers to the hazard scenarios.

Having said the former, we are not pointing out that seismic amplification due to the soil site condition might not occur. And due to the scope of our paper, we do not perform any site response analysis (1D, 2D, 3D) after having obtained the simple GMPE-based ground motions. Nonetheless, we agree with the reviewer that providing a clarification in this regard will benefit the quality of the paper. Accordingly, we have provided a note in the discussion section that will benefit the understanding of the assumptions within our procedure for this large-scale study. Thus, we have decided to include in the updated version of the paper a recent citation (Aguilar et al., 2019) that clearly describes the importance of addressing soil-amplification in Lima.

**2.5.**>> "Details of the bathymetric data for the case study are missing. These should be included in full since they have a direct effect in the outcomes of a tsunami scenario analysis".

Yes, the bathymetric and topographic data are absolutely essential for tsunami simulations. However in the given study we do not aim at the validation of the model

for a given scenario which would require the most adequate topography model available. Rather we investigate the tsunami impact for varying magnitudes and strive for a systematic comparison by varying only very few parameters. But of course it is a crucial point and following this comment we included more details on the data used for the simulations.

Accordingly we have updated the new version of the manuscript as follows: "The model bathymetry and topography were built from several data sets. The ocean part is based on the GEBCO bathymetry (General bathymetric chart of the ocean, GEBCO\_08 Grid, see http://www.gebco.net). The coastal topography is given by SRTM values (Shuttle radar topography mission, 30m resolution, see https://www2.jpl.nasa.gov/srtm/) whereas in the pilot area Lima/Callao additionally the measurements of the TanDEM-X mission (see (Krieger et al., 2007) given at a spatial resolution of 12m were provided to the RIESGOS consortium. In this region, the available data sets were combined to a joint product and augmented by nautical charts in shallow areas by the project partner EOMAP. All these data were bilinearly interpolated to the triangular mesh and slightly smoothed to allow for stable simulations".

2.6. >>"Section 3.5 should include, for a better understanding, a graphical distribution of the nodes (geocells) after using one or another aggregation scheme. Even if Table 1 shows some interesting information, it is not easy to imagine the changes from one to another. Something similar to what is shown in Figure 4b, but for each of the aggregation schemes.

We have been considering this suggestion carefully but providing independent printed figures for each resultant CVT-based for the other models (with 10,000 and 50,000 geocells) would not necessarily lead to a better visualization upon printed figures. It is already quite difficult to see independent cells in figure 4b (only 5,000 geocells). Considering the large size of the study area as well as the limited are for a printed figure, some geocells will be basically displayed as point clouds. We consider that the subset study areas that are provided in Fig. 10, 14, 15 may be useful enough to provide a visual comparison between the various models upon the actual printed version, and most importantly, along with their respective normalized loss metric.

Nevertheless, following your advice, we have decided to provide the data models and scripts that we have constructed during the elaboration of our study. These models are supplementary data to the paper. They are assigned an independent DOI, and are accordingly cited in the new version of the manuscript. We consider that this is a transparent approach that could also benefit future readers who will be able to better understand, reuse and cite these datasets. Examples can be found accessing the following revision links:

https://dataservices.gfzpotsdam.de/panmetaworks/review/f932840b5c130da18c3a9d407e85f086ce0874b80ed bd796e0f096ba94d89cc4/

**https://dataservices.gfzpotsdam.de/panmetaworks/review/0470cd1366982c5e319c5c39ca1c2e524b213d8f9b7 868c98aad804fceba33d0/**

**2.7.**

---

## Author Comment (AC2)

**Responses to Anonymous Referee #2**

Thank you for your helpful review. Please find our answers to each of your comments below.

**General comments:**

1. <<"*From the reviewer's point of view, both the Central Voronoi Tessellations (CVT) itself and the application are topics of great interest since this is a useful method related to earthquake and tsunami risk assessment for communities affected or potentially affected by these threat*".

   We sincerely thank you your positive comments.

2. <<"*However, some process details were not explained clearly. This article uses the scientific research results of several scholars to get the research results*".

   You are right, we test the impact and benefits of variable resolution building exposure models on risk assessment making use of some previous "scholars" models as inputs. However, inputs such as seismic ground motion fields and tsunami inundation maps (for six scenarios) were constructed by us through our own computational resources. For this aim, of course we had to consider existing methods (i.e. GMPE, spatial cross-correlation, and wave propagation models) to construct ours for the study area. This is a conventional approach in research and maximises the reproductively and added value of the methods proposed by others.

   Please note that the aforementioned inputs, along with the use of existing fragility models are always transversally used throughout the processing risk chain. The main degree of freedom in our modelling correspond to the presence of various customised CVT-based geocells. They are indeed based on well-known mathematical developments (i.e. Lloyd, 1982; Voronoï, 1908). We fully agree that the CVT have been used in other hazard related applications, such climatological modelling (e.g. Ju et al., 2011; Zarzycki and Jablonowski, 2014) but only recently for exposure modelling using seismic vulnerability classes (Pittore et al., 2020). We present for the first time the manner of how they can be constructed using underlying combinations of geospatial distributions and then achieve a larger resolution where it matters for risk assessment. We have updated the introduction in order to better frame our scope and developments from the existing models used in the study.

3. <<"How to verify the research results?"

   In the submitted version of the manuscript we clearly stated that we do not aim to verify the resultant scenario-based loss estimates for Lima. In fact, such a validation is practically impossible without actually having experienced such events in Lima. Such an issue is recently discussed by Tozato et al., (2021). When catastrophe modelling is done out of the area where the empirical fragility models used were originally calibrated (e.g. Japan or the Indic Ocean), or

based on the use of analytical fragility functions, then these studies are only presented to provide insights about the consequences of future risk scenarios.

Several studies have developed such strategy (i.e. Vera San Martín et al., 2018; Park et al., 2019) and following the discussions we had these last years with stakeholders both in Europe and South America such scenarios are useful for preparedness, planning as well as for a better understanding of the spatial distribution of the physical vulnerability of a city even if they cannot be verified.

Thorough the paper we also constantly compare our findings with similar features observed by other studies in order to check consistency of our results (e.g. Figueiredo and Martina, 2016; Markhvida et al., 2017; Park et al., 2017).

In the submitted text we had clearly stated in lines 596-600: *"(…) we are not claiming that the scenario-based economic losses we have presented for the residential building stock of Lima are completely exhaustive. Instead, through the adoption of the condition tree, we have drawn a branched methodological workflow to explore the differential impact of the exposure aggregation models, and the selection of building schemes on the epistemic and thematic uncertainties that are embedded in scenario-based risk applications".* We consider that keeping this sentence is prudent and necessary.

4. <<"What is the innovative idea or technology of this article?*

This work presents for the first time a first contribution to find an adequate balance in the resolution of the exposure model with the spatial resolution and variability of the hazard intensities for risk assessment. The necessity of this research topic had been already suggested by other experts in that area (i.e. by Douglas, 2007; Ordaz et al. 2019; Zuccaro et al. 2018). We found that CVT-based models adequate to be used in the aforementioned purpose. As explained in the paper, our method contrasts with the current state of the art related to building exposure modelling (aggregation) that neglects the variability of the hazard intensities in their derivation. Current approaches simply uses administrative boundaries for exposure aggregation and risk computation. Thus, we consider that developing this new paradigm and the subsequent sensitivity analyses performed are themselves innovative. Moreover through the development of the manuscript the reader can realise characteristics related to the CVT models such as being computationally more efficient for risk computation. Although the former aspects are extensively described in the Introduction section of the paper, we make sure to emphasise this novelty in the new version of the manuscript.

Moreover, we make use of the concept of inter-scheme conversion matrices to further prove their usefulness to derive exposure models (i.e. spatial distribution of building classes and replacement costs). This is novel because if we can know these characteristics for a single exposure scheme (e.g. seismic-oriented), we could get the same descriptors for another vulnerability scheme (e.g. tsunamis). This procedure ensures the comparability across the

different schemes and this compatibility had not been considered so far in the related scientific literature for multi-hazard exposure modelling. This aspect also outlines that various exposure models existing in the literature can actually be complemented and compared in a probabilistic manner.

Another innovative idea: we test the proposed method over the residential building portfolio of an important megacity that has been strongly affected during non-instrumental times by earthquakes, and tsunamis. We observe that the The importance of addressing such scenarios for Lima as well as the comparison of our results with the few existing scenarios for Lima may be of the interest of many readers interested by risk scenarios for this city.

5. *<<"The general comment for the whole paper is that the reviewer has not been able to find enough significant points regarding the principal criteria of the reviewing process".*

   We sincerely expect that after having provided the former clarifications, the reviewer now can visualise the positive impact of our study and its novelty.

**Specific comments:**

1. *<<" Section 1 (page 4): The authors could highlight the advantages of the CVT, (1) (2) (3)…*

   You are right. Following this comment we realise the need to provide since the beginning of the manuscript a brief description enumerating the advantages of the CVT-based models. Basic characteristics such as compactness, stability, contiguity are some of these features that have been mentioned more clearly and cited accordingly in the suggested part of the new paper.

   Nevertheless we would prefer to also keep another list of advantages that we presented in the Conclusion section (lines 566-582). In these lines we have enumerated the main advantages of CVT-models to spatially aggregate the exposure model, risk computations, efficiently, and spatial representability. We believe that these statements require proper justifications that are only achieved throughout the development of ideas provided in the other sections of the paper.

2. *<<" Section 3.2: A numerical calculations table is needed to show the spatial resolution, time step, spatial range, and what water depth and elevation data is used. What governing equations are used in TsunAWI. Some detail about TsunAWI should be introduced"*

   As suggested, we extended the section on tsunami modelling and included more information on the approach used in the study.

   Part of the modified manuscript:

   "The wave propagation and tsunami inundations are obtained through numerical simulations using the finite element model TsunAWI which employs a triangular mesh with variable resolution, allowing for a flexible way to discretize the model domain with good representation

of coastline and bathymetric features. Since the simulation of the inundation process needs high resolution, the mean mesh resolution given by the triangle edge length amounts to around 20m in the coastal area of Lima and Callao. TsunAWI is based on the nonlinear shallow water equations including parameterisations for bottom friction and viscosity. Table 1 summarizes some of the most important model quantities. The wetting and drying scheme is based on an extrapolation method projecting model quantities between the ocean part and the dry land part of the model domain".

**Table 1.** Summary of TsunAWI model parameters used in the tsunami simulations.

| Numerical approach | Time step/ Integration time | Resolution range (Triangle edge length) | Bottom friction parameterization | Viscosity parameterization |
|---|---|---|---|---|
| Finite Elements | 0.1sec / 4 hrs | From 6km (deep ocean) to 7m (coastal pilot areas) | Manning (n=0.02 constant value) | Smagorinsky |

The model bathymetry and topography were built from several data sets. The ocean part is based on the GEBCO bathymetry (General bathymetric chart of the ocean, GEBCO_08 Grid, see http://www.gebco.net). The coastal topography is given by SRTM values (Shuttle radar topography mission, 30m resolution, see https://www2.jpl.nasa.gov/srtm/ ) whereas in the pilot area Lima/Callao additionally the measurements of the TanDEM-X mission (Krieger et al., 2007) given at a spatial resolution of 12m were provided by the project partner DLR to the RIESGOS consortium. In this region the available data sets were combined to a joint product and augmented by nautical charts in shallow areas by the project partner EOMAP. All these data were bilinearly interpolated to the triangular mesh and slightly smoothed to allow for stable simulations."

3. <<" *Section 3.6: What are the advantages of Suppasri's method and De Risi's method? Which method is the last choice? This issue should be discussed.*

The Suppasri et al., (2013) and De Risi et al., (2017) schemes, comprise different building classes (see figure 6 in the submitted paper). For instance, Suppasri define three types of wooden building classes (W1, W2, W3) whilst De Risi only accounts a single wooden class. As explained, this is because the storey range is only addressed in the first one. Moreover these models were derived in a very different manner whose implications are described in our paper as well in each of them. They even have different validity ranges of tsunami inundation height (20 m versus 10 m respectively). They have different number of damage states (six in Suppasri, five in DeRisi (it omits the damage state # 1) (see table 4 of the respective publication). The latter was mentioned in lines 435- 437 in our submitted paper.

At a first glance the model of De Risi might be considered as a better modelling approach because of its more robust derivation through multinomial logistic regression and with similar values as if flow velocity was accounted. Increasingly meaningful research in tsunami fragility

should not rely on the material type as the only descriptor of the tsunami fragility, instead, building height and other attributes should be always addressed (Charvet et al., 2017). The aforementioned aggregation procedure imposed by the De Risi scheme (in terms of the storey ranges) can largely impact the results. A similar effect had already been described for seismic risk assessment by Crowley et al., (2005). Hence, the differences between their respective risk outcomes might not only due to the parameters that made up the fragility functions, but also due to the aggrupation of different building classes (Suppasri) into a less diversified one (de Risi). Due to these limitations, and crude adoptions of these models for a South American context (out of the calibrated area), we invited the reader to realise the importance of counting with locally calibrated exposure, fragility and financial consequence models (lines 534-536).

We therefore do not have a last choice of the selected model. We prefer to clearly inform the reader about the differences between the two models, show how these two models impact the final loss computation and use these two models to evaluate the epistemic uncertainty associated to this model choice.

Moreover, we kindly let you know that we have decided to provide the data models and scripts that we have constructed during the elaboration of our study. These models are supplementary data to the paper. They are assigned an independent DOI, and are accordingly cited in the new version of the manuscript. We consider that this is a transparent approach that could also benefit future readers who will be able to better understand, reuse and cite these datasets. Examples can be found accessing the following review links:

https://dataservices.gfz-potsdam.de/panmetaworks/review/f932840b5c130da18c3a9d407e85f086ce0874b80edbd796e0f096ba94d89cc4/

https://dataservices.gfz-potsdam.de/panmetaworks/review/0470cd1366982c5e319c5c39ca1c2e524b213d8f9b7868c98aad804fceba33d0/

We also let you know that we asked an editor (a native English speaker) to provide us a strict language review. The new version has been significantly been improved in that regard.

We sincerely thank the reviewer for the time invested in providing us the very constructive feedback and comments.

With best regards,

*The team of authors.*

**References**

Charvet, I., Macabuag, J., Rossetto, T., 2017. Estimating Tsunami-Induced Building Damage through Fragility Functions: Critical Review and Research Needs. Frontiers in Built Environment 3, 36. https://doi.org/10.3389/fbuil.2017.00036

Crowley, H., Bommer, J.J., Pinho, R., Bird, J., 2005. The impact of epistemic uncertainty on an earthquake loss model. Earthquake Engineering & Structural Dynamics 34, 1653–1685. https://doi.org/10.1002/eqe.498

De Risi, R., Goda, K., Yasuda, T., Mori, N., 2017. Is flow velocity important in tsunami empirical fragility modeling? Earth-Science Reviews 166, 64–82. https://doi.org/10.1016/j.earscirev.2016.12.015

Douglas, J., 2007. Physical vulnerability modelling in natural hazard risk assessment. Natural Hazards and Earth System Sciences 7, 283–288. https://doi.org/10.5194/nhess-7-283-2007

Figueiredo, R., Martina, M., 2016. Using open building data in the development of exposure data sets for catastrophe risk modelling. Natural Hazards and Earth System Sciences 16, 417–429. https://doi.org/10.5194/nhess-16-417-2016

Ju, L., Ringler, T., Gunzburger, M., 2011. Voronoi Tessellations and Their Application to Climate and Global Modeling, in: Lauritzen, P., Jablonowski, C., Taylor, M., Nair, R. (Eds.), Numerical Techniques for Global Atmospheric Models. Springer Berlin Heidelberg, Berlin, Heidelberg, pp. 313–342. https://doi.org/10.1007/978-3-642-11640-7_10

Krieger, G., Moreira, A., Fiedler, H., Hajnsek, I., Werner, M., Younis, M., Zink, M., 2007. TanDEM-X: A Satellite Formation for High-Resolution SAR Interferometry. IEEE Transactions on Geoscience and Remote Sensing 45, 3317–3341. https://doi.org/10.1109/TGRS.2007.900693

Lloyd, S., 1982. Least squares quantization in PCM. IEEE Transactions on Information Theory 28, 129–137. https://doi.org/10.1109/TIT.1982.1056489

Markhvida, M., Ceferino, L., Baker, J.W., 2017. Effect of ground motion correlation on regional seismic lossestimation: application to Lima, Peru using across-correlated principal component analysis model, in: SBN 978-3-903024-28-1. Presented at the Safety, Reliability, Risk, Resilience and Sustainability of Structures and Infrastructure. 12th Int. Conf. on Structural Safety and Reliability, Christian Bucher, Bruce R. Ellingwood, Dan M. Frangopol (Editors), Vienna, Austria.

Ordaz, M., Salgado-Gálvez Mario Andrés, Huerta Benjamín, Rodríguez Juan Carlos, Avelar Carlos, 2019. Considering the impacts of simultaneous perils: The challenges of integrating earthquake and tsunamigenic risk. Disaster Prevention and Management: An International Journal 28, 823–837. https://doi.org/10.1108/DPM-09-2019-0295

Park, H., Alam, M.S., Cox, D.T., Barbosa, A.R., Lindt, J.W. van de, 2019. Probabilistic seismic and tsunami damage analysis (PSTDA) of the Cascadia Subduction Zone applied to Seaside, Oregon. International Journal of Disaster Risk Reduction 35, 101076. https://doi.org/10.1016/j.ijdrr.2019.101076

Park, H., Cox, D.T., Barbosa, A.R., 2017. Comparison of inundation depth and momentum flux based fragilities for probabilistic tsunami damage assessment and uncertainty analysis. Coastal Engineering 122, 10–26. https://doi.org/10.1016/j.coastaleng.2017.01.008

Pittore, M., Haas, M., Silva, V., 2020. Variable resolution probabilistic modeling of residential exposure and vulnerability for risk applications. Earthquake Spectra 36, 321–344. https://doi.org/10.1177/8755293020951582

Suppasri, A., Mas, E., Charvet, I., Gunasekera, R., Imai, K., Fukutani, Y., Abe, Y., Imamura, F., 2013. Building damage characteristics based on surveyed data and fragility curves of the 2011 Great East Japan tsunami. Natural Hazards 66, 319–341. https://doi.org/10.1007/s11069-012-0487-8

Tozato, K., Takase, S., Moriguchi, S., Terada, K., Otake, Y., Fukutani, Y., Nojima, K., Sakuraba, M., Yokosu, H., 2021. Real-time Tsunami Force Prediction by Mode Decomposition-Based Surrogate Modeling. Natural Hazards and Earth System Sciences Discussions 2021, 1–36. https://doi.org/10.5194/nhess-2021-77

Vera San Martín, T., Rodriguez Rosado, G., Arreaga Vargas, P., Gutierrez, L., 2018. Population and building vulnerability assessment by possible worst-case tsunami scenarios in Salinas, Ecuador. Natural Hazards 93, 275–297. https://doi.org/10.1007/s11069-018-3300-5

Voronoï, G., 1908. Nouvelles applications des paramètres continus à la théorie des formes quadratiques. Premier mémoire. Sur quelques propriétés des formes quadratiques positives parfaites. Journal für die reine und angewandte Mathematik 1908, 97–102. https://doi.org/10.1515/crll.1908.133.97

Zarzycki, C.M., Jablonowski, C., 2014. A multidecadal simulation of Atlantic tropical cyclones using a variable-resolution global atmospheric general circulation model. Journal of Advances in Modeling Earth Systems 6, 805–828. https://doi.org/10.1002/2014MS000352

Zuccaro, G., De Gregorio, D., Leone, M.F., 2018. Theoretical model for cascading effects analyses. International Journal of Disaster Risk Reduction 30, 199–215. https://doi.org/10.1016/j.ijdrr.2018.04.019

---

## Author Comment (AC3)

**Responses to Anonymous Referee #3**

Thank you for your helpful review. Please find our answers to each of your comments below.

**General comment.**

<<"*This manuscript is about valuating risks for natural hazards, presenting a methodological proposal based on CVT. Case studies in peril suggest that the CVT-based variable-resolution exposure models proposed have a high efficiency in integrating large-area building portfolios to adequately estimate earthquake and tsunami losses The topic of the paper fits the Journal well. Overall, I suggest the manuscript can be accepted after minor revisions in the following aspect:*

*This study is interesting, but in the revised version, the authors are suggested to provide more explanation about the novel points of this manuscript. Indeed I may miss something, but a clearer clarification may be better*"

Thank you for your nice comments and your suggestions. It is very meaningful for us that you have perfectly highlighted the main goal of our study while at the same finding it interesting.

This work presents for the first time a first contribution to find an adequate balance in the resolution of the exposure model with the spatial resolution and variability of the hazard intensities for risk assessment. The necessity of this research topic had been already suggested by other experts in that area (i.e. by Douglas, 2007; Ordaz et al. 2019; Zuccaro et al. 2018). We found that CVT-based models adequate to be used in the aforementioned purpose. As explained in the paper, our method contrasts with the current state of the art related to building exposure modelling (aggregation) that neglects the variability of the hazard intensities in their derivation. Current approaches simply uses administrative boundaries for exposure aggregation and risk computation. Thus, we consider that developing this new paradigm and the subsequent sensitivity analyses performed are themselves innovative.

Although the former aspects are extensively described in the Introduction section of the paper, we make sure to emphasise this novelty in the new version of the manuscript. Thus, we also have made a clearer link between the current state of the art regard and the conclusion chapter. This list enumerates more exhaustively the main advantages of CVT-based models over the conventional approaches for exposure modelling and aggregation (administrative aggregation entities).

Moreover, we make use of the concept of inter-scheme conversion matrices to further prove their usefulness to derive exposure models (i.e. spatial distribution of building classes and replacement costs). This is novel because if we can know these characteristics for a single exposure scheme (e.g. seismic-oriented), we could get the same descriptors for another vulnerability scheme (e.g. tsunamis). This procedure ensures the comparability across the different schemes and this compatibility had not been considered so far in the related scientific literature for multi-hazard exposure modelling. This aspect also outlines that various exposure models existing in the literature can actually be complemented and compared in a probabilistic manner.

Another innovative idea: we test the proposed method over the residential building portfolio of an important megacity that has been strongly affected during non-instrumental times by earthquakes, and tsunamis. The importance of addressing such scenarios for Lima as well as the comparison of our results with the few existing scenarios for Lima may be of the interest of many readers interested by risk scenarios for this city.

Through this response we would also like to let you know that we have decided to provide the data models and scripts that we have constructed during the elaboration of our study. These models are supplementary data to the paper. They are assigned an independent DOI, and are accordingly cited in the new version of the manuscript. We consider that this is a transparent approach that could also benefit future readers who will be able to better understand, reuse and cite these datasets. An example can be found accessing the following link:

https://dataservices.gfz-potsdam.de/panmetaworks/review/f932840b5c130da18c3a9d407e85f086ce0874b80edbd796e0f096ba94d89cc4/

We also let you know that we asked an editor (a native English speaker) to provide us a strict language review. The new version has been significantly been improved in that regard.

We sincerely thank the reviewer for the time invested in providing us the very constructive feedback and comments.

With best regards,

*The team of authors.*

**References**

Douglas, J., 2007. Physical vulnerability modelling in natural hazard risk assessment. Natural Hazards and Earth System Sciences 7, 283–288. https://doi.org/10.5194/nhess-7-283-2007

Ordaz, M., Salgado-Gálvez Mario Andrés, Huerta Benjamín, Rodríguez Juan Carlos, Avelar Carlos, 2019. Considering the impacts of simultaneous perils: The challenges of integrating earthquake and tsunamigenic risk. Disaster Prevention and Management: An International Journal 28, 823–837. https://doi.org/10.1108/DPM-09-2019-0295

Zuccaro, G., De Gregorio, D., Leone, M.F., 2018. Theoretical model for cascading effects analyses. International Journal of Disaster Risk Reduction 30, 199–215. https://doi.org/10.1016/j.ijdrr.2018.04.019

---

## Author Comment (AC4)

**Responses to Anonymous Referee #4**

Thank you for your helpful review. Please find our answers to each of your comments below.

**General comments:**

1. *<<"The study is useful as the method have a high efficiency in estimating the earthquake and tsunami loss".*

   We sincerely thank you your positive feedback.

2. *<<"However, there are some details in the technical part should be explained clearly by the author, like the process of select earthquake scenario described in the line 290, " A Mw 9.0 tsunami scenario was selected among a catalogue...", but I cannot find the description of the catalogue, (I'm sorry if I miss something)".*

   We are sorry about the confusion caused by the reference to the catalogue that had not been introduced before. It actually refers to the tsunami scenario database that was calculated within the RIESGOS project. From the corresponding state of this database the 'worst-case' event with regard to the inundation extend in Lima/Callao was chosen for the design of the focus map. The database is still growing and in future studies we will refine the 'worst-case' method by taking into account aggregates of a series of scenarios. Consequently, we have removed from the revised manuscript the sentence referring to 1,000 offshore scenarios.

   With regard to the tsunami modelling, in this study we do not aim at a validation of inundation results for some given event and corresponding realistic source model. Rather we investigate the tsunami impact for events characterized by a range of magnitudes and, having a systematic comparison in mind, we aim at the variation of only one parameter, namely the slip value. The sources used in the study are based on the historic event from 1746 and the source area is taken from Jimenez et al., (2013).

   The scaling law is not specified in Jimenez et al., (2013) but the paper addresses a magnitude range of Mw 8.6 to Mw 9.0, the rigidity is specified to a value of $mu= 4.5x10^{10} N/m^2$ and the source dimension is set to 550x140 km, separated into five subfaults. By modifying the source area, we would increase the number of degrees of freedom considerably since the local bathymetry affects the evolution of the extended initial sea surface elevation in a nonlinear way. Therefore, we restricted in this specific study the source modification to the slip value (constant among subfaults, the values ranging from 2.73m for Mw 8.5 to 15.36m for Mw 9.0), thus scaling the total energy of the event but keeping the other quantities constant.

   The model bathymetry and topography were built from several data sets. The ocean part is based on the GEBCO bathymetry (General bathymetric chart of the ocean, GEBCO_08 Grid, see http://www.gebco.net). The coastal topography is given by SRTM values (Shuttle radar topography mission, 30m resolution, see https://www2.jpl.nasa.gov/srtm/ ) whereas in the pilot area Lima/Callao additionally the measurements of the TanDEM-X mission (Krieger et al., 2007)

given at a spatial resolution of 12m were provided by the project partner DLR to the RIESGOS consortium. In this region the available data sets were combined to a joint product and augmented by nautical charts in shallow areas by the project partner EOMAP. All these data were bilinearly interpolated to the triangular mesh and slightly smoothed to allow for stable simulations."

As suggested, we extended the section on tsunami modelling and included more information on the approach used in the study.

Part of the modified manuscript:

*"The wave propagation and tsunami inundations are obtained through numerical simulations using the finite element model TsunAWI which employs a triangular mesh with variable resolution, allowing for a flexible way to discretize the model domain with good representation of coastline and bathymetric features. Since the simulation of the inundation process needs high resolution, the mean mesh resolution given by the triangle edge length amounts to around 20m in the coastal area of Lima and Callao. TsunAWI is based on the nonlinear shallow water equations including parameterisations for bottom friction and viscosity. Table 1 summarizes some of the most important model quantities. The wetting and drying scheme is based on an extrapolation method projecting model quantities between the ocean part and the dry land part of the model domain."*

Table 1. Summary of TsunAWI model parameters used in the tsunami simulations.

| Numerical approach | Time step/ Integration time | Resolution range (Triangle edge length) | Bottom friction parameterization | Viscosity parameterization |
|---|---|---|---|---|
| Finite Elements | 0.1sec / 4 hrs | From 6km (deep ocean) to 7m (coastal pilot areas) | Manning (n=0.02 constant value) | Smagorinsky |

Through this response we would also like to let you know that we have decided to provide the data models and scripts that we have constructed during the elaboration of our study. These models (including the tsunami inundation maps) constitute five data repositories and are supplementary data to the paper. They are assigned an independent DOI, and are accordingly cited in the new version of the manuscript. We consider that this is a transparent approach that could also benefit future readers who will be able to better understand, reuse and cite these datasets. An example can be found accessing the following revision link:

https://dataservices.gfz-potsdam.de/panmetaworks/review/f932840b5c130da18c3a9d407e85f086ce0874b80edbd796e0f096ba94d89cc4/

We also let you know that we asked an editor (a native English speaker) to provide us a strict language review. The new version has been significantly been improved in that regard.

We sincerely thank the reviewer for the time invested in providing us the very constructive feedback and comments.

With best regards,

*The team of authors.*

**References**

Jimenez, C., Moggiano, N., Mas, E., Adriano, B., Koshimura, S., Fujii, Y., Yanagisawa,  and H., 2013. Seismic Source of 1746 Callao Earthquake from Tsunami Numerical Modeling. Journal of Disaster Research 8, 266–273. https://doi.org/10.20965/jdr.2013.p0266

Krieger, G., Moreira, A., Fiedler, H., Hajnsek, I., Werner, M., Younis, M., Zink, M., 2007. TanDEM-X: A Satellite Formation for High-Resolution SAR Interferometry. IEEE Transactions on Geoscience and Remote Sensing 45, 3317–3341. https://doi.org/10.1109/TGRS.2007.900693

---

## Author Response (AR1)

Dear editor,

Thank you for your corrections to our paper.

- We have integrated the suggestions by the 4 reviewers following our responses to each of them. In the revised version of the manuscript we are highlighting such updates.
- References to data repositories and software (with their brand new DOI) are also cited. An example of a review link is provided herein:

  https://dataservices.gfz-potsdam.de/panmetaworks/review/032b84c9eb546155d1070387cd93cdd70fedf6185634552e93154f7a19b16e61/

- We have improved the quality of the figures. Some of them have a different numeration in this version to ensure a more coherent reading.
- We also let you know that we asked a native English speaker to provide us a strict language review. The new version has been significantly been improved in that regard.

We sincerely thank you for your time,

With best regards,

*Juan Camilo Gomez-Zapata on behalf of the team of authors.*

---

## Author Response (AR2)

Dear editor,

Thank you for the suggestions to improve our paper.

Please find below a point-by point answer to your seven comments:

1. I fully agree with you that the paper is long. This is because it gathers a significant amount of work that started in 2018 where not only my personal research activities are explained, but also of my colleagues. Therefore, due to the very short time we got to revise the manuscript, I cannot reduce considerably the text of the paper without being fully approved once again by the other six coauthors. Also, considering that all the content written in the last submitted version was already approved by my colleagues after several months of checking it out, I doubt they will want to reduce their own contributions.

   Personally I consider that the manner the introduction and method parts are presented make the paper pedagogic enough to explain the paradigms we are tracking on. Moreover, please note that our paper contains many references and large figures (lots of them of the size of one page, which logically increase the amount of pages. The final count of words is 15,727, an average number for a complete research paper. This version contains only 1,565 words more (around one page of full content) in comparison with the first submitted version in March 98, 2021, a low number considering it already has all the required content by four reviewers. Having said that, I will be sincerely thankful if you allow us keeping the parts where we present the introduction, method and results as they were presented in the last submitted version. Thanks for your understanding.

2. Following your advice I have considerably reduced the Conclusion section. As suggested, I have moved some parts to the Discussion part. It really makes sense. I deeply thank you for this helpful recommendation. The relocated parts are highlighted in the author's track-changes file.

3. Indeed, in the answer provided to Dr. Mario Salgado we provide one figure (Figure 1 in that response text) as requested by him. Thus, this was just for illustrative purposes. Please note that in that answer we provide a complete explanation about the reasons why these comparison, in our opinion, does not make much sense. Thereby, we clearly wrote the folliong: *"Therefore, we consider that it would not very meaningful to the reader if we provide a graphical comparison of these two fragility models. Accordingly, in the journal paper we would prefer to avoid doing so"*. Thus, that figure was not provided in the revised version of the manuscript. Since Dr. Mario Salgado did not make any comment in this regard, we consider that our suggestion was accepted.

The minor issues listed as comments **4, 5** and **7** regarding the modification of figures, and the units in one table (comment **6**) have been successfully solved in the updated version of the manuscript.

We sincerely thank you for your time. With best regards,

Juan Camilo Gomez-Zapata on behalf of the team of authors

---

## Author Response (AR3)

Dear NHESS editorial office,

I thank you for your feedback. As suggested, have updated the following changes to the updated manuscript:

- I have included the same phrase in the caption of Figure 6: "Map data: ©Google Earth 2021" to the caption of Figure 5.
- I have included the suggested statement to the caption of Figures 2 and 3.
- I have made sure that my name, Juan Camilo Gomez Zapata appears as the corresponding author both in the manuscript records as well as in the pdf.

Thanks a lot for your time.

With deep gratitude,

Juan Camilo Gomez Zapata